# A lung-selective delivery of mRNA encoding broadly neutralizing antibody against SARS-CoV-2 infection

Wanbo Tai [1,2,7], Kai Yang[3,7], Yubin Liu [1,7], Ruofan Li[4,7], Shengyong Feng[1,7], Benjie Chai[1,7], Xinyu Zhuang[5,7], Shaolong Qi[3], Huicheng Shi[1], Zhida Liu[6], Jiaqi Lei[3], Enhao Ma[1], Weixiao Wang[1,2], Chongyu Tian[1,2], Ting Le[1,2], Jinyong Wang[2], Yunfeng Chen [2], Mingyao Tian [5] ✉, Ye Xiang [4] ✉, Guocan Yu [3] ✉ & Gong Cheng [1,3] ✉

The respiratory system, especially the lung, is the key site of pathological injury induced by SARS-CoV-2 infection. Given the low feasibility of targeted delivery of antibodies into the lungs by intravenous administration and the short half-life period of antibodies in the lungs by intranasal or aerosolized immunization, mRNA encoding broadly neutralizing antibodies with lung-targeting capability can perfectly provide high-titer antibodies in lungs to prevent the SARS-CoV-2 infection. Here, we firstly identify a human mono-clonal antibody, 8-9D, with broad neutralizing potency against SARS-CoV-2 variants. The neutralization mechanism of this antibody is explained by the structural characteristics of 8-9D Fabs in complex with the Omicron BA.5 spike. In addition, we evaluate the efficacy of 8-9D using a safe and robust mRNA delivery platform and compare the performance of 8-9D when its mRNA is and is not selectively delivered to the lungs. The lung-selective delivery of the 8-9D mRNA enables the expression of neutralizing antibodies in the lungs which blocks the invasion of the virus, thus effectively protecting female K18-hACE2 transgenic mice from challenge with the Beta or Omicron BA.1 variant. Our work underscores the potential application of lung-selective mRNA antibodies in the prevention and treatment of infections caused by circulating SARS-CoV-2 variants.

Severe acute respiratory syndrome coronavirus 2 (SARS-CoV-2) is the etiological agent of the coronavirus disease 2019 (COVID-19) pandemic, which has caused the hospitalization and death of millions of individuals worldwide[1]. There have already been numerous attempts to use vaccines developed by multiple technical routes to target the SARS-CoV-2 spike protein that has already been explored for clinical usage to prevent COVID-19[2–6]. In addition, neutralizing monoclonal antibodies (mAbs) have exhibited potential remarkable promise for

[1]New Cornerstone Science Laboratory, Tsinghua-Peking Joint Center for Life Sciences, School of Medicine, Tsinghua University, Beijing 100084, China. [2]Institute of Infectious Diseases, Shenzhen Bay Laboratory, Shenzhen 518132, China. [3]Key Laboratory of Bioorganic Phosphorus Chemistry & Chemical Biology, Department of Chemistry, Tsinghua University, Beijing 100084, China. [4]Beijing Advanced Innovation Center for Structural Biology, Beijing Frontier Research Center for Biological Structure, Center for Infectious Disease Research, School of Medicine, Tsinghua University, Beijing 100084, China. [5]Changchun Veterinary Research Institute, Chinese Academy of Agricultural Sciences, Changchun 130122, China. [6]Shanxi Academy of Advanced Research and Innovation, Taiyuan 030032, China. [7]These authors contributed equally: Wanbo Tai, Kai Yang, Yubin Liu, Ruofan Li, Shengyong Feng, Benjie Chai, Xinyu Zhuang. ✉e-mail: klwklw@126.com; yxiang@mail.tsinghua.edu.cn; guocanyu@mail.tsinghua.edu.cn; gongcheng@mail.tsinghua.edu.cn

COVID-19 treatment[7–11]. To date, several therapeutic anti-SARS-CoV-2 mAbs have been licensed for use in humans[12–14]. The prevention of emerging infectious diseases, such as COVID-19, by antibody treatment involves several advantages; in contrast to vaccines that may require several weeks or even months to achieve protective effects, passive immunization by administration of antibodies shows the potential for a near-immediate onset of action[15–17]. Nonetheless, the clinical application of antibody treatment is largely hampered by the high cost of development and manufacturing[18]. The applications are also restricted by the inability to target the tissue of interest and the short half-life. Thus, the development of an approach that may deliver antibodies toward targeted tissues with high effectiveness and low cost will revolutionize the feasibility of using antibody therapy and prophylaxis for COVID-19 and other infectious diseases in a widespread setting.

Messenger RNA (mRNA)-based biotechnology has been developed for prophylactic and therapeutic strategies to combat infectious diseases, e.g., vaccination development[19–26], protein replacement therapy[27,28], and CRISPR/Cas nuclease-based genetic editing[29,30] against pathogenic infections. The US Food and Drug Administration (FDA) recently authorized two mRNA vaccines enabled by lipid nanoparticles (LNPs) against COVID-19 for emergency use, which represented a key milestone in the application of mRNA therapeutics. Aside from COVID-19, multiple mRNA vaccine candidates against influenza viruses[31,32], respiratory syncytial virus[33], and rabies virus[34] have also been developed and are currently applied in human clinical trials. In contrast to comprehensive immune stimulation by systemic administration of mRNA-encoding antigens, the clinical success of mRNA-based antibody therapeutics is largely reliant on the development of safe, efficient, and highly selective delivery systems; these systems transport mRNA toward specific tissues and subsequently produce the desired therapeutic effect and minimize systemic toxicity[35]. Indeed, the majority of mRNA administered by traditional LNP systems targets the liver after systemic administration[36]. Selectively delivering mRNA toward specific organs in vivo remains a major challenge in the clinical application of mRNA-based therapeutics. As a typical respiratory-transmitted pathogen, SARS-CoV-2 prophylactically targets the human lungs for infection[37,38]. The in situ production of anti-SARS-CoV-2 neutralizing antibodies in the lungs increases the antibody concentration in the targeted organ, thus quickly achieving the necessary concentration to achieve therapeutic effects. Benefiting from the rapid antibody response, this strategy is especially promising for prophylaxis in an emergency to immediately avoid the possible outbreak. Moreover, the rapidly increasing neutralizing antibodies in the lungs is conducive to the remission of disease, presenting a therapeutic alternative after infection. In this study, we developed lung-selective LNPs that delivered mRNAs encoding a broadly neutralizing antibody, to the mouse lungs in a highly efficient manner; as a results, a remarkable therapeutic effect was achieved in the prevention and treatment of infection by SARS-CoV-2 variants.

## Results

### Isolation and characterization of SARS-CoV-2-specific antibody 8-9D

We screened plasma samples from a cohort of inactivated vaccine (BBIBP-CorV)-immunized subjects for the presence of neutralizing antibodies against SARS-CoV-2. Six individuals with the highest plasma neutralizing titers were selected to isolate receptor-binding domain (RBD)-specific memory B cells by fluorescence-activated cell sorting (FACS) (Fig. 1a and Supplementary Fig. 1). We obtained 118 RBD-specific monoclonal antibodies (mAbs), of which 20 mAbs exhibited potent neutralizing activity against SARS-CoV-2[39]. In addition, antibody 8-9D was obtained from a donor with broad neutralizing activity against SARS-CoV-2 variants (Supplementary Table 2 and Supplementary Table 3).

We measured the binding kinetics of 8-9D using a biolayer interferometry (BLI) binding assay. The dissociation constant ($K_D$) of 8-9D with the wild-type SARS-CoV-2 RBD was 5.43 nM, and the binding of 8-9D to the RBD of Omicron BA.2 and BA.4/BA.5 was retained and partially diminished by BA.1 (Fig. 1b). We next assessed the antibody competition with human angiotensin-converting enzyme 2 (hACE2), which revealed that 8-9D blocked the binding of RBD and hACE2 (Fig. 1c). We further determined the footprint of antibody 8-9D on the RBD by analyzing the epitope specificity of 4 structurally defined mAbs, CB6 (Class 1), C121 (Class 2), COV2-2130 (Class 3), and COVA1-16 (Class 4), with BLI. The competition profile indicated that 8-9D recognized an epitope overlapping with the footprints of Class 1, Class 2, and Class 4 antibodies but not Class 3 antibodies (Fig. 1d).

Next, we investigated the inhibitory ability of 8-9D against SARS-CoV-2 infection in vitro. 8-9D exhibited potent neutralizing activity against wild-type and all previously circulating variants of concern (VOCs) including Alpha, Beta, Gamma, and Delta as well as previously circulating variants of interest (VOIs) including Lambda, Mu, Kappa, Eta, Iota v1, Iota v2, Epsilon, and Zeta in pseudotyped virus assay (Fig. 1e, f). In addition, 8-9D was capable of neutralizing Omicron subvariants (BA.1, BA.2, BA.2.12.1, BA.3, BA.4, BA.4.6, BA.5, BF.7, and XD) (Fig. 1g). Using a cytopathic effect (CPE) inhibition assay, we further measured the inhibitory activity of 8-9D against live SARS-CoV-2. In line with the data from the pseudovirus assay, 8-9D potently neutralized authentic SARS-CoV-2 wild type, Beta, Delta with $IC_{50}$ values of 22 to 46 ng/mL and was impaired by Omicron BA.1 and BA.2 (Fig. 1h).

### Structural basis of broad-spectrum neutralization by 8-9D

We then determined the cryo-EM structure of 8-9D Fab in complex with the ectodomain of the spike from Omicron BA.5 in order to understand how 8-9D neutralizes the virus. Two major conformations were obtained after performing a three-dimensional classification of the particles. In one of its conformations, the spike exhibits two "up" RBDs, each of them binds an 8-9D Fab. In the alternative conformation, the spike contains all RBDs in the "up" conformation and three bound 8-9D Fabs (Supplementary Fig. 2). The spike with all "up" RBDs was reconstructed at a global resolution of 3.0 Å (Supplementary Fig. 3b). However, the local resolution at the interface in which 8-9D Fab binding is poor and the map is not sufficient for ab initial model building (Supplementary Fig. 3c). We thus performed further local refinement on the RBD as well as the bound 8-9D Fab, and improved the local resolution at the contact interface to 3.32 Å (Supplementary Fig. 3b). The map from the local refinements is sufficient for the initial model building (Fig. 2a).

Each 8-9D Fab binds the RBD with a large buried surface of 1140 Å², of which ~60% is contributed by the heavy chain, while ~40% is contributed by the light chain. A total of 24 RBD residues have direct contacts with the 8-9D Fab (Supplementary Table 4). 23 residues in 8-9D Fab are involved in direct contact. Among these, 16 residues are from the heavy chain, while 7 residues are from the light chain. Between these contact residues, 21 hydrogen bonds are formed and 5 pairs of hydrogen bonds are found with a distance equal to or less than 2.7 Å, including (RBD-T415$^{OG1}$-HC-S57$^{OG}$, RBD-Y421$^{OH}$-HC-G55$^N$, RBD-L455$^O$-HC-Y34$^{OH}$, RBD-Q493$^{NE2}$-HC-S101$^O$, RBD-S494$^O$-LC-Y32$^{OH}$) (Fig. 2d). Some residues form multiple pairs of hydrogen bonds with 8-9D Fab, including N417, S494, N477 and N487. Among these, N417 and S494 mediate 3 pairs and N477 and N487 mediate 2 pairs of hydrogen bonds (Fig. 2d). These strong hydrogen bond networks play a major role in the interaction between the 8-9D Fab and the RBD, which possesses multiple mutations at the contacting surface in different variants.

There is a significant overlap between the epitope recognized by 8-9D and the surface recognized by ACE2. 8-9D covers ~70% of the ACE2 binding site. The bound angle of 8-9D is approximately 30° away from that of ACE2 (Fig. 2b, c). According to the epitope, 8-9D can be

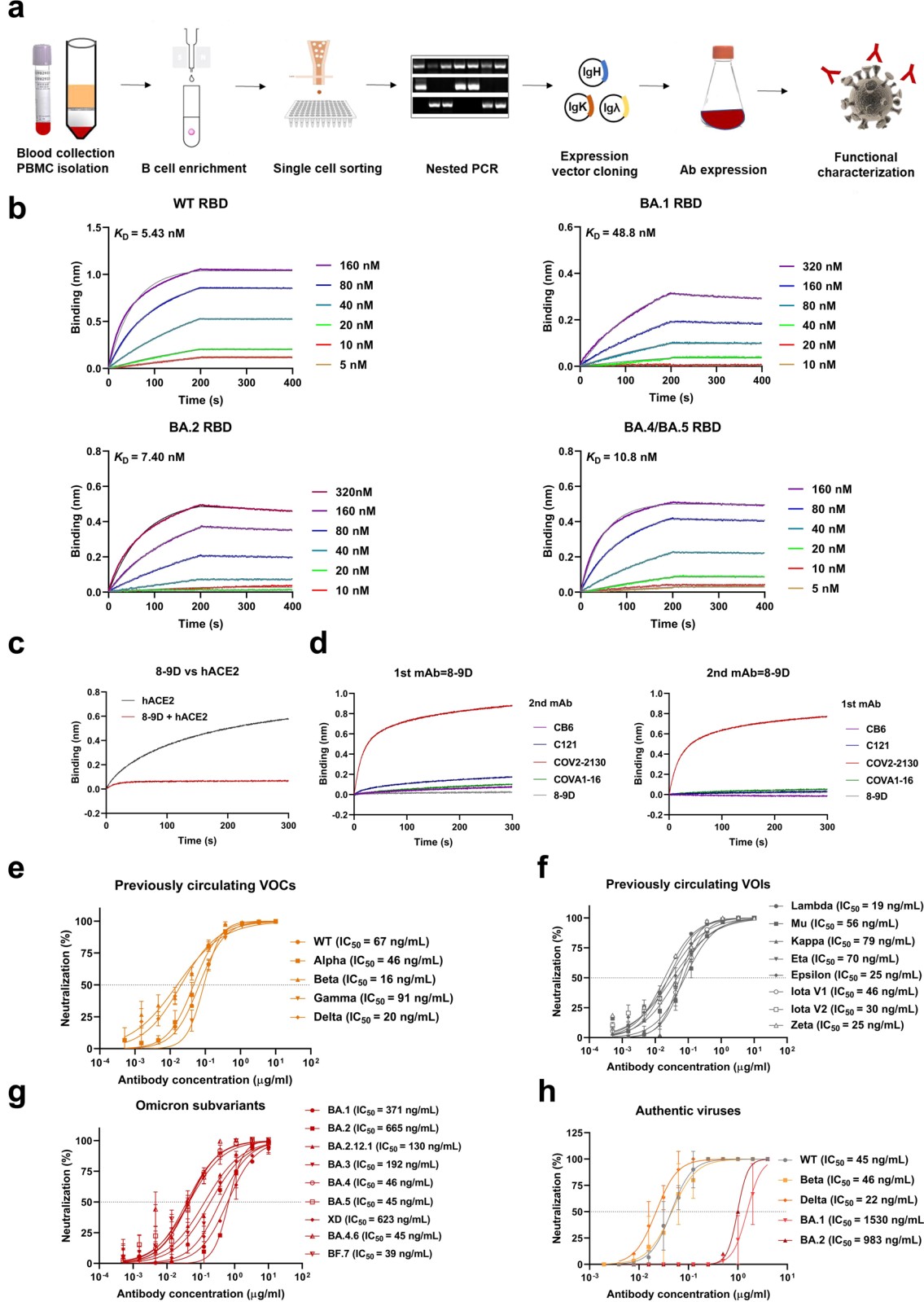

categorized into Class I neutralization antibodies that directly compete with ACE2 for binding RBD[40] (Fig. 2b).

In viral neutralization assays, 8-9D showed strong neutralizing activity against most SARS-CoV-2 variants, including Alpha, Beta, Gamma, Delta, Lambda, Mu, Kappa, Eta, Iota, Epsilon, Zeta, and Omicron BA.4/5 (Fig. 1e–h). Most SARS-CoV-2 variants, including the Omicron subvariants BA.1, BA.2, BA.2.12.1, BA.3, BA.4, BA.4.6, BA.5,

BF.7, and XD, have multiple mutations located in the epitope of 8-9D. The structural details of the epitope can be used to interpret the broad-spectrum neutralizing activity of 8-9D. Compared with the RBD of the original SARS-CoV-2 strain, the omicron strain BA.5 has 6 mutations in the contact surface with 8-9D, including D405N, K417N, S477N, Q498R, N501Y, and Y505H. The complex structure shows that five of the six mutated residues form hydrogen bonds with 8-9D (Fig. 2e). Structural

**Fig. 1 | Characterization of SARS-CoV-2 targeting monoclonal antibody 8-9D.**
**a** Schematic for the neutralizing monoclonal antibody selection and evaluation.
**b** Binding affinity of 8-9D to RBDs of WT, Omicron BA.1, Omicron BA.2 and Omicron BA.4/BA.5 by biolayer interferometry (BLI). The association and dissociation of the response curves of 8-9D are shown. The gray lines represent the fitted curves based on the experimental data. Equilibrium dissociation constants (KD) are shown above each plot. **c** Antibody 8-9D and hACE2 competition for binding to SARS-CoV-2 RBD determined by BLI. The traces show the binding of the hACE2 to preformed 8-9D-RBD complexes. The level of reduction in shift of 8-9D + hACE2 compared to the hACE2-only control indicates the blocking capacity of the antibody. **d** Epitope competition measured by BLI. The antibody 8-9D was assayed for epitope specificity with 4 structurally defined monoclonal antibodies, CB6 (class 1), C121 (class 2), COV2-2130 (class 3), and COVA1-16 (class 4). The traces represent the binding of the

second antibody (2nd Ab) to preformed first antibody (1st Ab)-RBD complexes in an in-tandem binning assay. **e**–**g** Neutralization of monoclonal antibody 8-9D against pseudotyped SARS-CoV-2 wild type, previously circulating variants of concern (VOCs) (**e**), previously circulating variants of interest (VOIs) (**f**), and Omicron subvariants (**g**). The 50% inhibitory concentration (IC$_{50}$) values are shown. The dashed line indicates a 50% reduction in viral infectivity. The curves were fitted by nonlinear regression (log [inhibitor] vs. normalized response, variable slope), $n = 2$ biologically independent samples, and the results are representative of 2 or more independent experiments with similar results. Data are mean ± SEM of the experiments. **h** Neutralizing activity of 8-9D against authentic SARS-CoV-2 wild type, Beta, Delta, Omicron BA.1, and Omicron BA.2 variants as in (**e**–**g**). Source data are provided as a Source data file.

modeling of the residues in the original strain shows that D405, Q498, and N501 can form hydrogen bonds with S93, S67, and I29 of the 8-9D light chain, respectively. In the Omicron strain BA.5, similar hydrogen bonds can be maintained by the mutations D405N, Q498R, and N501Y (RBD-N405$^{ND2}$-LC-S93$^{OG}$, distance: 2.9 Å; RBD-R498$^{NH1}$-LC-S31$^{OG}$, distance: 3.0 Å; and RBD-Y501$^{OH}$-LC-S31$^{OG}$, distance: 3.2 Å). Similarly, structural modeling shows that K417 can form a hydrogen bond with Y53 of the 8-9D heavy chain and that S477 can form hydrogen bonds with T29 of the 8-9D heavy chain. In the Omicron strain BA.5, the mutation K417N leads to the establishment of two more hydrogen bonds through the side chain of the asparagine residue with both residue Y53 and residue Y34 of the 8-9D heavy chain (RBD-N417$^{ND2}$-HC-Y34$^{OH}$, distance: 3.2 Å; RBD-N417$^{ND2}$-HC-Y53$^{OH}$, distance: 3.7 Å). Apart from the hydrogen bond between the main chain amino group of S477 from the RBD and T29 from the 8-9D heavy chain (RBD-S477$^{N}$-HC-T29$^{OG1}$, distance: 3.7 Å), the S477N mutation also leads to the establishment of a new hydrogen bond between N477 and T29 of the 8-9D heavy chain (RBD-S477$^{OD1}$-HC-T29$^{OG1}$, distance: 3.3 Å). The establishment of new hydrogen bonds through the mutations should improve the affinity between RBD and the 8-9D heavy chain.

In our structural modeling, Y505 in the wild-type strain can form a hydrogen bond with H90 of the 8-9D light chain. Van der Waals interactions with H90 of the 8-9D light chain are still present despite the disruption of hydrogen bond by Y505H (Fig. 2e). Since the van der Waal interactions are inadequate to compensate for the loss of hydrogen bonds, the mutation Y505H in BA.5 should decrease the interaction between the RBD and 8-9D. However, since 24 residues of the RBD are involved in the direct contact with 8-9D, the binding energy loss caused by Y505H may be compensated by other residues, such as residues N417 and N477. To confirm this hypothesis, we determined the binding of 8-9D with wild-type RBD and wild-type RBD that bears the Y505H mutation. The dissociation constants determined by BLI indicated that the Y505H RBD ($K_D = 5.60$ nM) binds mAb 8-9D with a similar affinity as that of the wild-type RBD ($K_D = 5.79$ nM) (Supplementary Fig. 4a). Mutation Q493R emerges in BA.1, BA.2, and BA.3 subvariants and is also located in the 8-9D epitope. Similarly, BLI analysis showed that the Q493R mutation does not impair the binding affinity of 8-9D to RBD (Supplementary Fig. 4a). In simulated Q493R RBD binding with 8-9D, the hydrogen bond (RBD-Q493$^{NE2}$-HC-S101$^{O}$, distance: 2.3 Å) is disrupted and a new hydrogen bond can form between R493 of RBD and S101 of the 8-9D heavy chain (RBD-R493$^{NE}$-HC-S101$^{O}$, distance: 2.9 Å) (Supplementary Fig. 4b). In the interaction of Q493R-RBD and 8-9D, the local affinity around R493 should decrease due to the increased distance between RBD-R493$^{NE}$-HC-S101$^{O}$.

### Structural comparisons with other broadly neutralizing antibodies

To date, some broadly neutralizing antibodies against Alpha, Beta, Delta strains and a few broadly neutralizing antibodies against Omicron strains have been reported. Many of these reported antibodies can be categorized as Class I RBD-targeting antibodies that target the

ACE2 binding site, such as CB6[9], B38[41], S2K146[42], and F61[43]. These antibodies share similar binding orientations on the RBD and all exhibit high neutralizing activities against the original virus strain (IC$_{50}$: 8-9D, 67 ng/mL; S2K146, 16 ng/mL; F61, 6 ng/mL; CB6, 36 ng/mL; B38, 177 ng/mL). All these antibodies exhibit a high binding affinity to the wild-type RBD and have an epitope of over 1000 Å$^2$ (8-9D, 1140 Å$^2$; S2K146, 979 Å$^2$; F61, 1145 Å$^2$; CB6, 1088 Å$^2$; B38, 1211 Å$^2$). Their epitopes also cover the hydrophobic pocket (around residue F456) on the neck of the RBD, which is conserved among widespread VOCs (Supplementary Fig. 5a, b, right panel). Compared with these reported broadly neutralizing antibodies, 8-9D targets similar common features, including a large epitope, a strong binding affinity, targeting the conservative hydrophobic pocket and the inclusiveness to multiple mutations inside the epitope by the establishment of new interactions.

In recent epidemic variants, including Alpha, Beta, Delta strains and especially the Omicron subvariants, mutated hotspots such as S371L, S373P (Omicron), L452R (Delta, Kappa and Epsilon), and E484K/A (Beta, Gamma, Zeta and Omicron), are located outside the epitopes of these broadly neutralizing antibodies, which should exert little influence on the antibody-RBD interaction. Although 8-9D, CB6, and B38 share similar epitopes (Supplementary Fig. 4a, b), their neutralizing activities are unequal against variants with the same mutations within their epitopes. It has been reported that CB6 and B38 are sensitive to the K417N mutation identified in Beta and Omicron strains[44,45]. The K417N mutation disrupts the interaction between the RBD and CB6 or B38, and unlike 8-9D, CB6, and B38 cannot generate new hydrogen bonds with N417, which helps to maintain the 8-9D-RBD affinity. Similar to 8-9D, S2K146 shows broad neutralization activity against Omicron subvariants BA.1-3 (IC$_{50}$: ~16 ng/mL) with reduced efficacy against BA.4/5 (IC$_{50}$: 221 ng/mL)[46] and F61 neutralizes Omicron BA.1-4 strains with high efficiency (IC$_{50}$: ~12 ng/mL). S2K146 can bind to key ACE2 contact positions L455, F486, Q493, Q498, and N501 and should form electrostatic interactions with K417N, Q493R, and N501Y substitutions in Omicron subvariants[42]. F61 binds with Omicron BA.1 RBD with high affinity ($K_D = 5$ nM) and forms extensive hydrogen bonds and salt bridges with N417, N477, K478, R493, and H505 substitutions in Omicron subvariants[43].

### Characterization of the lung-selective LNP mRNA delivery system compared to systematic-selective LNPs

Lung-selective mRNA encoding broadly neutralizing antibodies can directly induce protection against SARS-CoV-2 in the lungs. One of the key factors that regulate the biodistribution of LNPs is the surface charge[47,48]. Different surface charges result in the distinction of serum proteins adsorbed to LNPs, which determines the subsequent organ-specific distribution of LNPs[49]. Specifically, positively charged LNPs predominantly accumulate in the lungs of mice, while near-neutral LNPs present a liver accumulation. Therefore, the manipulation of surface charge by LNP formulation emerges as a feasible way to achieve the organ-selective delivery of LNPs. The systematic-selective lipid nanoparticle (Liver-LNPs) system is a classic four-component formulation

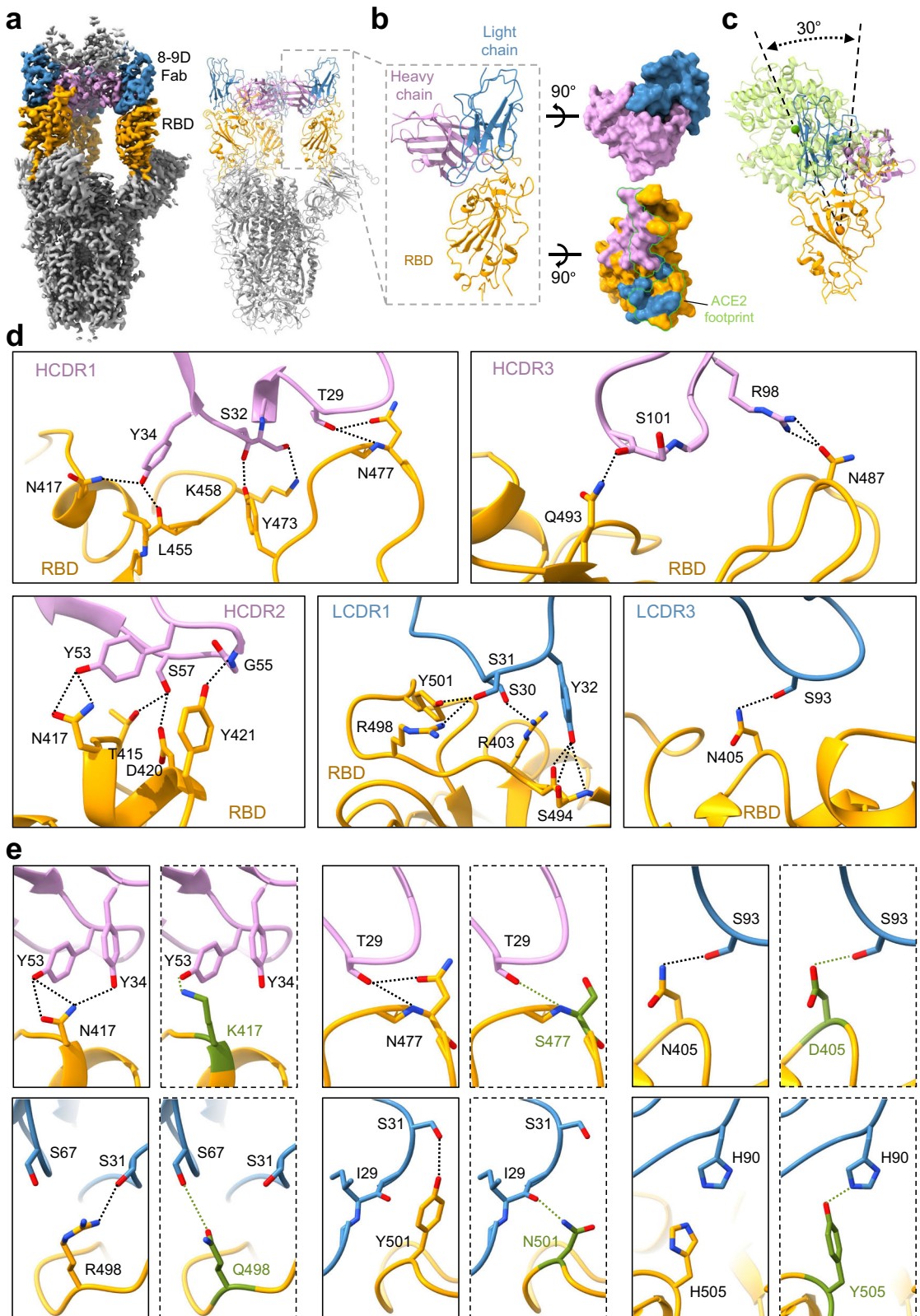

that consists of cholesterol, 1,2-distearoyl-sn-glycero-3-phospho-choline (DSPC), poly(ethylene glycol) lipids (PEG-lipid), and ionizable lipids (Fig. 3b). The ionizable lipid can be converted into its quaternary ammonium salt form via a simple and effective reaction with methyl iodide to afford a cationic targeting lipid (Fig. 3a). By adding the additional permanently cationic targeting lipid and replacing DSPC with 1,2-dioleoyl-sn-glycero-3-phosphoethanolamine (DOPE) in the Liver-LNP

formulation (Fig. 3b), we successfully obtained lung-selective LNPs (Lung-LNPs), which were capable of precisely delivering mRNA to the lungs. mRNA-loaded LNPs were formed by mixing an aqueous buffer solution containing mRNA and a lipid ethanol solution at the aqueous-to-ethanol ratio of 3/1, *v/v*. Firefly luciferase mRNA (mRNA$^{Luci}$) was encapsulated by the two organ-selective LNPs, giving Liver-LNPs@mRNA$^{Luci}$ and Lung-LNPs@mRNA$^{Luci}$. Both the obtained

**Fig. 2 | Structural characterization of monoclonal antibody 8-9D. a** Surface rendered representations and ribbon diagrams showing the structure of the Omicron BA.5 spike in complex with three 8-9D Fabs. Only the variable region of 8-9D Fab is displayed in the ribbon diagrams (right). The "up" RBDs are in orange, the heavy chain of 8-9D is colored pink, and the light chain is colored blue. The selected region in the dashed square shows one RBD with one bound Fab. **b** Ribbon diagrams showing the RBD interacting with the variable region of the 8-9D Fab (left). The rendered surface of 8-9D Fab and RBD in an open-up view (right). The epitopes on the RBD recognized by the 8-9D heavy chain and light chain are colored pink and blue, respectively. The ribbons of 8-9D Fab and RBD are colored the same as in (**a**). The ACE2 footprint on RBD is highlighted with green lines. **c** Ribbon diagrams showing the comparisons of the binding orientations of ACE2 and 8-9D on the RBD.

The 8-9D Fab and RBD are colored the same as in (**a**) and ACE2 is colored green. The green, pink, and orange balls are the centroids of ACE2, 8-9D Fab, and RBD, respectively. **d** Detailed ribbon and stick diagrams showing the hydrogen bonds between the RBD and complementarity-determining region (CDR) of 8-9D heavy and light chains. RBD, 8-9D heavy and light chains are colored the same as in (**a**). Residues involved in hydrogen bond formation are shown in sticks with oxygen and nitrogen atoms colored red and blue, respectively. **e** Ribbon and stick diagrams comparing the corresponding residues in Omicron BA.5 RBD (in a solid rectangle) and in the wild-type RBD (in a dashed rectangle). The residues and hydrogen bonds in wild type RBD are colored green. The complex structure of 8-9D and the wild-type RBD was obtained through structural modeling with COOT[70] and Foldit Standalone[74]. RBD, 8-9D heavy and light chains are colored the same as in (**a**).

Liver-LNPs@mRNA$^{Luci}$ and Lung-LNPs@mRNA$^{Luci}$ were uniformly spherical assemblies with a diameter of ~60–100 nm according to the transmission electron microscopy (TEM) images (Fig. 3c). Dynamic light scattering (DLS) analysis exhibited consistent results, presenting average size of 80.1 ± 15.9 nm for Liver-LNPs@mRNA$^{Luci}$ and 90.2 ± 16.0 nm for Lung-LNPs@mRNA$^{Luci}$ (Fig. 3d, e). Furthermore, following 48 h of incubation in phosphate buffer solution (PBS), neither of these two organ-selective LNPs showed any noticeable alterations in average diameter, indicating promising stability in the physiological environment (Fig. 3f). Notably, the zeta potential of Liver-LNPs@mRNA$^{Luci}$ was measured to be −11.3 ± 8.8 mV, while the Lung-LNPs@mRNA$^{Luci}$ presented a positive charge of 25.0 ± 9.5 mV, which might explain the different organ distribution preferences[49] (Fig. 3g, h).

The organ-selective delivery and in vivo expression in organs were well illustrated by in vivo and ex vivo imaging. After receiving Liver-LNP@mRNA$^{Luci}$ i.v. administration for 2 h, strong fluorescence signals were observed in the mouse abdomen, and the fluorescence lasted for at least 24 h. Mice treated with Lung-LNPs@mRNA$^{Luci}$ presented fluorescence signals in the chest, suggesting a different organ-selective delivery compared to the Liver-LNPs@mRNA$^{Luci}$ (Fig. 3i). Excitingly, ex vivo imaging revealed conclusive evidence of specific organ-selective capabilities of Liver-LNPs and Lung-LNPs. Figure 3j demonstrates the successful liver-selective delivery and mRNA expression of Liver-LNPs@mRNA$^{Luci}$. Notably, the Lung-LNPs@mRNA$^{Luci}$ demonstrated selective lung-selective delivery and high expression level of mRNA$^{Luci}$ at 2 h post i.v. injection, without observed signals from other organs including the liver (Fig. 3k). The statistical fluorescence results from both in vivo and ex vivo imaging for the two organ-selective LNPs are shown in Fig. 3l. In vivo imaging indicated the high luminescence signals 2 h post injection and the expression of mRNA$^{Luci}$ could continue for up to 24 h. Ex vivo organ imaging revealed that the expression level of mRNA peaked at approximately 4 h post treatment, for both organ-selective LNPs. To further investigate the cell types in the livers or lungs that took up LNPs and translated mRNA into certain proteins, mice were injected with mRNA encoding enhanced green fluorescent protein (mRNA$^{eGFP}$) using Liver-LNPs and Lung-LNPs, and the lungs and livers were then collected 4 h later. Confocal laser scanning microscopy (CLSM) images explicitly revealed the high expression level of eGFP principally positioned in parenchymal hepatic cells and alveolar epithelial cells after treatment with Liver-LNPs@mRNA$^{eGFP}$ and Lung-LNPs@mRNA$^{eGFP}$, respectively (Fig. 3m).

The results of liver and renal function indices, including alanine aminotransferase (ALT), aspartate aminotransferase (AST), uric acid (UA), blood urea nitrogen (BUN), and creatinine (CREA) from mice at 6 h post administration of Liver-LNPs@mRNA$^{Luci}$ or Lung-LNPs@mRNA$^{Luci}$ showed no apparent differences compared to those of normal mice, suggesting that no hepatic or renal damage occurred during the treatment (Supplementary Fig. 6).

### Characterization of 8-9D mRNA-LNPs in vitro and in vivo

Based on the aforementioned excellent neutralizing potency of the 8-9D mAb, which was discovered using B cells from individuals

immunized with inactivated COVID-19 vaccines, we next developed 8-9D into an mRNA platform to reduce the cost and foster clinical application.

To construct and synthesize the 8-9D mRNA against SARS-CoV-2, optimized 5′-UTR and 3′-UTR, poly A tail (104 bp) and Cap1 were added to the mRNA sequence for the stability and high expression efficacy of mRNA. In addition, a Flag tag and V5 tag were included at the carboxy terminus of the heavy chain and light chain, respectively (Supplementary Fig. 7). SDS–PAGE and Western blot analysis revealed the strong in vitro expression in HEK293T cells transfected with this construct (Supplementary Fig. 8a). Similarly, the translational efficacy of the 8-9D antibody was confirmed by the heavy chain and light chain specific immunofluorescence staining (Supplementary Fig. 8b).

Following the in vitro assays, we evaluated the delivery efficiency of the 8-9D mRNA in C57BL/6 mice after i.v. administration of the mRNA packaged with the above-described Lung-LNPs or Liver-LNPs. C57BL/6 mice were intravenously injected with a single administration of the 8-9D-encoded mRNA at a dose of 0.25 mg/kg. After the mRNA infusion, we first analyzed the resulting antibody expression from the Liver-LNPs@mRNA$^{8-9D}$ or Lung-LNPs@mRNA$^{8-9D}$ in sera, bronchoalveolar lavage fluid (BALF), lungs, and livers by ELISA (Fig. 4a). It was found that at 24 h after infusion, significantly higher titers of the 8-9D antibody were detected in the BALF and lungs of mice in the Lung-LNPs group than those in the Liver-LNPs group; as expected, the antibody levels in sera and livers were much higher after treatment with the Liver-LNPs system (Fig. 4b–e). We next determined the expression kinetics of the 8-9D antibody in BALF and lungs from the mRNA packaged with the Liver-LNPs or Lung-LNPs system (Fig. 4f,g). Benefiting from the high stability and translation capability of circle RNA, we were able to extend the half-life of the antibody in the lungs and lung alveolar lavage fluid to nearly two weeks by encapsulating circle RNA encoding 8-9D (Supplementary Fig. 9), which demonstrated that this delivery platform possessed promising potentials in combating SARS-CoV-2 and other respiratory infectious diseases. Specific 8-9D mRNA expressed cells types in lungs could may affect the antibody's anti-infection efficacy. After the injection of Lung-LNPs@mRNA$^{eGFP}$, the analysis of eGFP-expressing cell types in the lungs using flow cytometry revealed a transient significant increase in endothelial cell expression at 2 h post injection (Supplementary Fig. 10a), and by the 6 h mark, the expression levels across various cell types tended to reach a balance among epithelial cells, endothelial cells and lymphocytes (Supplementary Fig. 10b). In addition, we also screened for inflammatory signals that might be triggered by the lipid formulas and the mRNA, such as immune cell activation and cytokine production. The infiltrations of the specific immune cell populations that we evaluated were not changed significantly in the lungs at 6 h post injection by the Liver-LNPs@mRNA$^{8-9D}$ or Lung-LNPs@mRNA$^{8-9D}$ (Supplementary Fig. 11). In the panel of thirteen cytokines and chemokines related to inflammatory responses (Supplementary Fig. 12), we detected higher concentrations of IL-6, IL12P70, IL-17A, and IL-10 in the blood samples at 6 h post injection

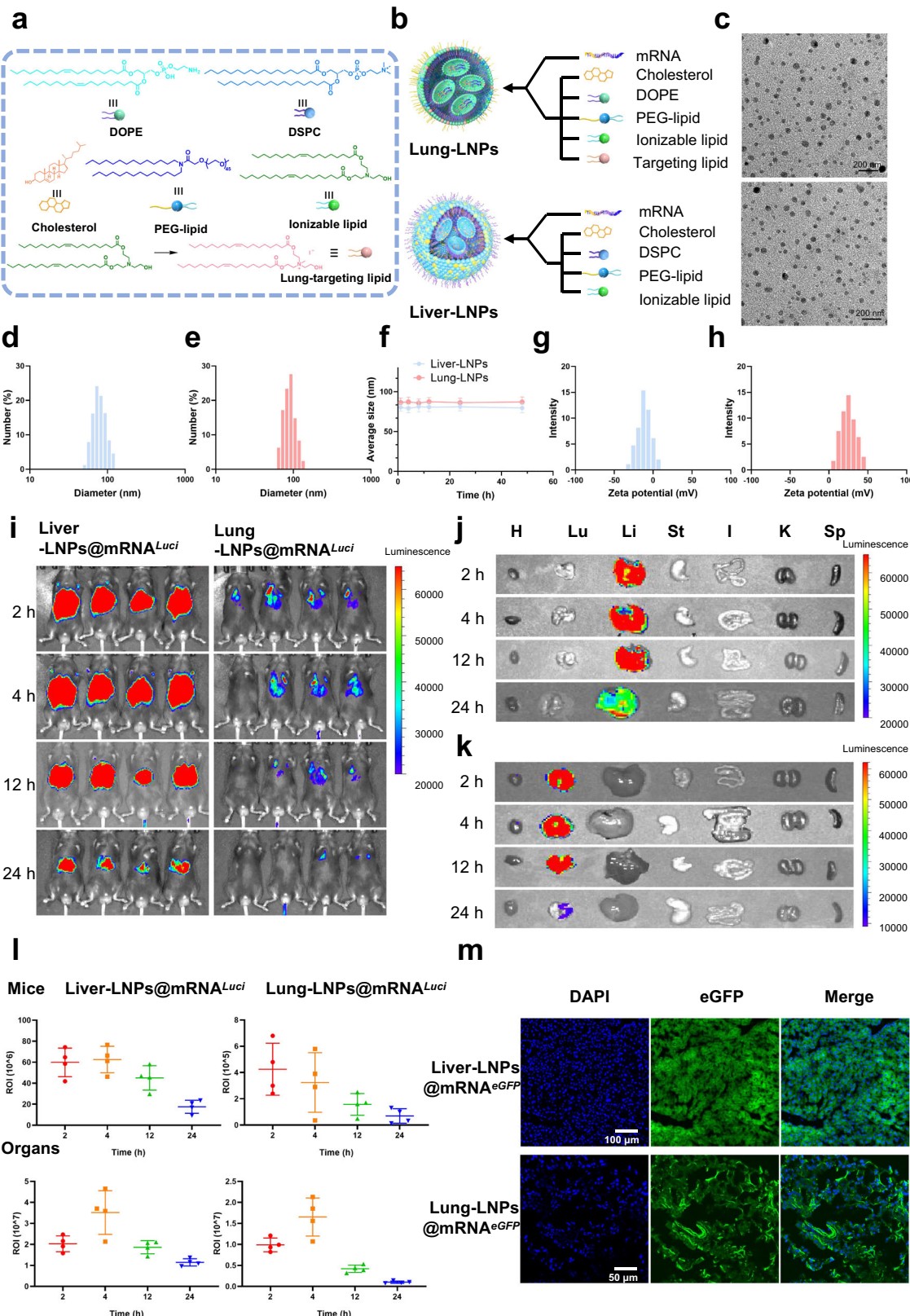

of Liver-LNPs@mRNA$^{8-9D}$ than PBS. In contrast, only IL-10 increased significantly in the Lung-LNPs@mRNA$^{8-9D}$ group. Other cytokines and chemokines were not significantly different from those in the PBS group. In addition, we measured the levels of three key cytokines including IL-6, TNF-α, and IL-1β at 1, 2, 4, and 8 d post injection of Lung-LNPs@mRNA$^{8-9D}$ (Fig. 4h–j), which confirmed the safety of this formulation. Moreover, the relatively positive charge of Lung-targeting LNPs may potentially interact with red blood cells. We conducted hemolysis experiments by incubating red blood cells with Lung-LNPs@mRNA$^{8-9D}$, of which negligible changes in hemolysis were monitored, further verifying its in vivo biocompatibility (Supplementary Fig. 13)[50,51]. These results indicated that the lung-selective 8-9D mRNA delivery system did not lead to a severe inflammatory response in the lungs, obviating the safety concerns.

**Fig. 3 | Characterization of organ-selective lipid nanoparticle systems.**
**a** Structural formula and schematic diagram of each component of lung-selective and liver-selective LNPs. **b** Schematic diagram of lung-selective and liver-selective LNPs. **c** Transmission electron microscopy (TEM) results: upper image (Lung-LNPs) and lower image (Liver-LNPs). **d**, **e** Dynamic light scattering (DLS) results for liver-selective (**d**) and lung-selective LNPs (**e**). **f** Particle size change for two particles in PBS, data are shown as the mean ± SD, $n = 3$ biologically independent samples. **g**, **h** Zeta potential analysis of liver-selective (**g**) and lung-selective LNPs (**h**), $n = 3$ independent experiments. **i**, In vivo imaging of liver-selective and lung-selective LNPs@mRNA$^{Luci}$ at different time points, $n = 4$ for each LNPs@mRNA$^{Luci}$ at different time points. **j**, **k** Visceral imaging of liver-selective and lung-selective LNPs@mRNA$^{Luci}$. The organs are heart (h), lung (Lu), liver (Li), stomach (St), intestine (I), kidney (K), and spleen (Sp). **l** The statistical results of luminescence intensity from mice inoculated with liver-selective and lung-selective LNPs@mRNA$^{Luci}$ at different time points, the ROI of liver-selective LNP statistics was the liver, and the ROI of lung-selective LNP statistics was the lungs, data are shown as the mean ± SD, $n = 4$ independent samples. **m** Imaging liver tissue or lung tissue of mice inoculated with liver-selective and lung-selective LNPs@mRNA$^{eGFP}$ at 24 h post injection. Source data are provided as a Source data file.

## Protection of the SARS-CoV-2 mouse model against challenge by selective delivery of 8-9D mRNA

The protective effect conferred by the lung-selective delivery of the mRNA encoding 8-9D was then assessed. To achieve this, K18-hACE2 mice were treated with a single i.v. administration of lung-selective LNPs (Lung-LNPs@mRNA$^{8-9D}$) or systematic-selective LNPs (Liver-LNPs@mRNA$^{8-9D}$) at a dose of 0.25 mg/kg 24 h before the challenge with the SARS-CoV-2 Beta variant at $2 \times 10^4$ TCID$_{50}$ or Omicron BA.2 at $2 \times 10^4$ TCID$_{50}$; this was performed to evaluate viral replication and pathology. The treated mice were euthanized on day 4 after the challenge to determine the tracheal and pulmonary virus levels using qRT–PCR (Fig. 5a). The analysis showed that the 8-9D mRNA delivered by the lung-selective and systemically selective LNP systems was associated with a statistically significant reduction in the viral loads in the lungs and trachea compared to that of the mock-treated group. Furthermore, the Lung-LNPs@mRNA$^{8-9D}$ exhibited nearly complete protection against the Beta variant infection and 100% protection from the Omicron BA.2 strain challenge (Fig. 5b, Supplementary Fig. 14a). We also demonstrated that the lung-selective mRNA delivery resulted in the absence of pulmonary damage, as determined by histopathologic evaluation (Fig. 5c, d, Supplementary Fig. 14b, c). To further demonstrate the effectiveness of targeted antibody delivery for post infection applications, we included protective studies involving administration after infection (Fig. 5e). The results revealed that administration of Lung-LNPs@mRNA$^{8-9D}$ 24 h after being infected with the coronavirus could also provide substantial protection, resulting in a protection rate approaching 100% (Fig. 5f–h, Supplementary Fig. 14d–f). Collectively, these results suggested that prophylaxis with 8-9D mRNA delivered with lung-selective LNPs can considerably relieve the disease induced by SARS-CoV-2 infection in a mouse model.

## Discussion

SARS-CoV-2 and its Omicron variants have infected over 770 million individuals to date[52]. To combat the COVID-19 pandemic, many therapeutic mAbs have been approved or obtained emergency use authorization by the US FDA or conditional marketing authorization by the European Union's European Medicines Agency for prevention and treatment[53,54]. Although these antibody therapeutics have shown promise in preventing or treating COVID-19 in clinical application, many challenges remain for implementing these therapies; the challenges mainly include a high cost of antibody production and a low effectiveness of mRNA delivery toward specific organs. In this study, we isolated a human monoclonal antibody named 8-9D from the B cells of an individual immunized by an inactive COVID-19 vaccine, which comprehensively presented potent neutralizing activity against circulating VOCs.

We successfully determined the cryo-EM structure of the Omicron BA.5 spike in complex with 8-9D Fabs at a 3.0 Å resolution. mAb 8-9D can be categorized as a Class I RBD-targeting antibody and prevents ACE2 from binding to the RBD with high affinity. The structural details of the epitope can be used to interpret the broad-spectrum neutralizing activity of 8-9D against most of the SARS-CoV-2 variants, including the Omicron subvariants BA.3, BA.4, and BA.5, which have multiple mutations located in the RBD epitope. 8-9D can accommodate most of the mutated hotspots, such as K417N, S477N, and N501Y. Due to the large buried surface, a single site mutation that breaks one hydrogen bond causes little effect on 8-9D binding to the RBD. However, in Omicron BA.1/2 subvariants, the occurrence of both Y505H and Q493R mutations should further weaken the interaction between RBD and 8-9D, which explains the result of relatively higher IC$_{50}$ of 8-9D neutralizing BA.1 and BA.2 subvariants.

Indeed, the clinical therapeutics of respiratory infectious diseases benefit greatly from the lung-selective in situ production of neutralizing mAbs, including the following: (1) the in situ production of neutralizing mAbs provides a wide variety of post translational modifications of antibodies, including glycosylation, deamidation, incomplete disulfide bond formation, etc. of which the modifications may offer the higher effectiveness and more stability of antibodies in in vivo circumstances[55]; and (2) lung-selective mRNA delivery can lead to a high dose of neutralizing antibodies through in situ production, which is necessary for prophylaxis or treatment of respiratory infection. In contrast, the systemic delivery route may not deliver the administered mRNAs into the tissue of interest and retain the materials in circulation[56,57]. In addition, the in situ production of neutralizing antibodies might result in prolonged and higher peak titers of the antibody compared with that of the protein format, because mRNAs can generate protein expression for a few days. (3) The mRNA-based in situ production of neutralizing mAbs shows a high safety to host cells. mRNAs are cannot access the nucleus of target cells, which reduces the risk of integration with the host genome that has been reported with adeno-associated DNA-based viral vectors[58,59]. (4) Compared to the equivalent protein therapeutics, the cost of lung-selective approach may be less. Since mRNAs can be directly delivered toward targeting tissues with susceptibility to infection, a much lower dose of mRNA is needed for LNP preparation. Thus, the in situ production of antibodies by mRNA-based technology circumvents potential problems of purification and heterogeneity of antibody production[60–62]. Taken together, these benefits underline the merit of mRNA-based lung-specific strategies in passive immunization against SARS-CoV-2 infection and other respiratory diseases.

Our lung-selective LNP formulation presents a safe and efficient solution to the organ-selective mRNA delivery system. The hydrolyzable ester group that comprises the ionizable lipid and targeting lipid backbones could be easily degraded by esterase in cells, thus decreasing the risk of long-term immunotoxicity of these LNPs. More importantly, the unsaturated fatty chains on both lipid backbones were flexible, resulting in comparatively small LNPs with diameters smaller than 100 nm and improved stability. The positive charge and small average size worked together to transport LNPs specifically to lungs with no occurrence of pulmonary embolism. The Lung-LNPs system can be readily obtained via the incorporation of cationic lipids with well-developed Liver-LNPs, ensuring both efficacy and safety. This strategy holds considerable potential for extension to other lung-selective LNPs, offering a broad range of applicability. Excitingly, the targeting capability, translation efficiency, and safety of Lung-LNPs can be further improved through the optimization of the metabolic/degradation rate as well as usage ratio of cationic lipids and the surface charge of Lung-LNPs. For example, lipid scaffolds made from

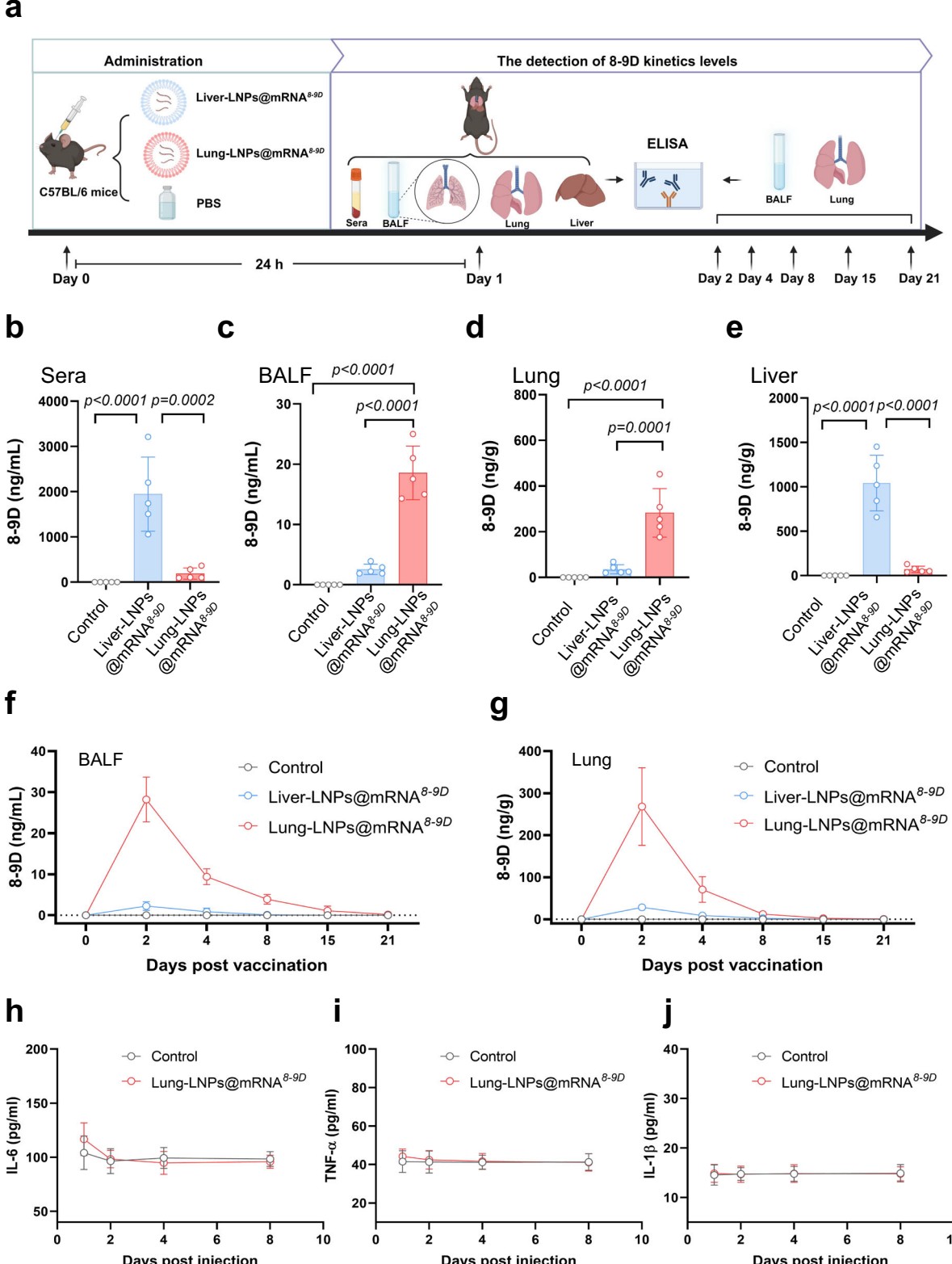

**Fig. 4 | In vivo activity and protective efficacy of 8-9D mRNA. a** Schematic for distribution and kinetics analysis of 8-9D mRNA after injection. C57BL/6 mice were i.v. injected with 5 μg of 8-9D mRNA with Lung-LNPs or Liver-LNPs as the delivery system. The animals (*n* = 5) were euthanized at 24 h post injection. **b**–**e** The sera (**b**), BALF (**c**), livers (**d**), and lungs (**e**) were collected for 8-9D detection by ELISA. 8-9D kinetics levels in BALF (**f**) and lungs (**g**) were further measured for the samples (*n* = 3) collected on day 2, 4, 8, 15, and 21 post injection. IL-6, TNF-α and IL-1β levels in sera at 1 day, 2 days, 4 days, and 8 days post injection of Lung-LNPs@mRNA^8-9D (**h**–**j**). Data are presented as mean ± SD and representative of two independent experiments with similar results. *P* values were determined by one-way ANOVA with Tukey's multiple comparison post-hoc test. Source data are provided as a Source data file.

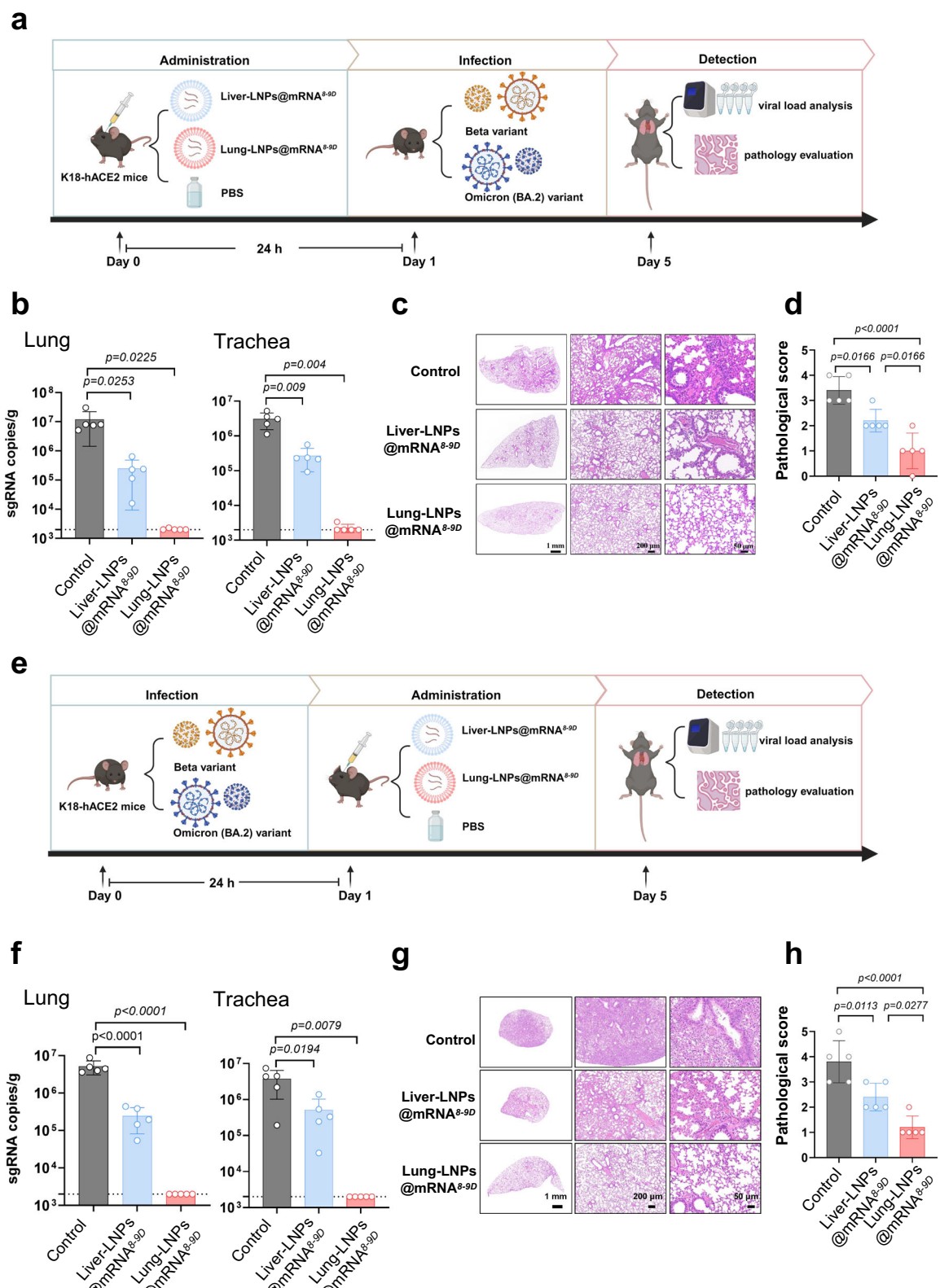

hydrolyzable structures such as carbonate ester and Schiff bases may enable the rapid degradation of cationic lipids after treatment, hence improving safety. In addition, the introduction of multiple quaternary ammonium sites allows efficient targeting and transfection with a minimal quantity of cationic lipids, resulting in reduced dosage and improved safety.

This study suggests that administering lung-selective LNP formulations containing mRNAs that encodes a broad neutralizing antibody to animals for passive immunization offers a prophylactic strategy against infection by SARS-CoV-2 variants. The lung-selective mRNA delivery of neutralizing antibodies provides great potential for the quick response of emerging SARS-CoV-2 variants. It is worth

**Fig. 5 | 8-9D mRNA protective efficacy in SARS-CoV-2 animal model. a–d** 8-9D mRNA protective efficacy as prophylaxis, the K18-hACE2 mice (*n* = 5) were i.v. injected with 5 μg of Liver-LNPs@mRNA*[8-9D]* or Lung-LNPs@mRNA*[8-9D]*. Twenty-four hours post LNP injection, the mice were challenged with 2 × 10⁴ TCID₅₀ Beta variant. Four days post challenge, the mice were euthanized, and the lungs and tracheae were collected for viral load analysis by qRT–PCR, *n* = 5 biologically independent animals (**b**). The lungs were fixed for pathology evaluation (**c**), and pathological scores (**d**) were determined for significant comparison. PBS-injected mice were set as control, *n* = 5 biologically independent animals. **e–h** 8-9D mRNA protective efficacy as treatment. The mice were first anesthetized and intranasally inoculated with 2 × 10⁴ TCID₅₀ of authentic SARS-CoV-2 Beta variant, and after 24 h, the mice were intravenously administered one dose of 5 μg of Liver-LNPs@mRNA*[8-9D]* or Lung-LNPs@mRNA*[8-9D]*, or PBS alone as a control. The mice were euthanized 4 days post infection to harvest lung tissues and trachea tissues for viral load test (**f**) or histopathology evaluation (**g**, **h**). Data are presented as mean ± SD and representative of two independent experiments with similar results, *n* = 5 biologically independent animals. *P* values were determined by one-way ANOVA with Tukey's multiple comparison post-hoc test. Source data are provided as a Source data file.

mentioning that the Lung-LNPs system exhibits versatility in its administration routes, including aerosol inhalation and intranasal inoculation, which enhance patient compliance and convenience in medication administration. These alternative routes can effectively target the lungs, facilitating the production of neutralizing antibodies against the COVID-19 virus. Furthermore, nebulization/intranasal vaccination using Lung-LNPs can also deliver antigen-encoding mRNAs, which elicit strong mucosal immunity, holding significant promise and practical benefits for the prevention and treatment of several diseases, such as COVID-19 and respiratory syncytial virus. Our forthcoming research endeavors will focus on augmenting the targeting, efficacy, and safety of Lung-LNPs, with a concerted effort towards advancing the clinical application of this delivery system.

In addition, the method exhibits versatility and potential for developing universal antibody-based therapeutics against other emerging and reemerging respiratory viruses in the future.

## Methods

### Ethnic statement
The protocol for blood collection was approved by the Institutional Review Board of Tsinghua University (Approval number: 20210061), and all participants provided written informed consent. All animal care and experimental procedures were approved by the Institutional Animal Care and Use Committee (IACUC) of Shenzhen Bay Laboratory (BACG202101), and Changchun Veterinary Research Institute of Chinese Academy of Agricultural Sciences (11-2022-032) in accordance with the relevant guidelines for the protection of animal subjects.

### Participants and blood sample processing
A cohort of healthy volunteers who were planning to immunized with COVID-19 vaccine was recruited for blood donation. These volunteers received two doses of the BBIBP-CorV (or Covilo) vaccine (4 weeks apart) and were boosted with the same vaccine at a median of 9 months after the second dose. Participant blood samples were collected over a time course covering before and after immunization to characterize the immune responses as previously described[39]. Participant blood samples with high binding and neutralizing activity against SARS-CoV-2 were selected for RBD-specific single B-cell sorting.

Blood samples were collected in K2 EDTA tubes (BD Vacutainer, Cat. No# 367525) and processed within 12 h. Briefly, the whole blood was centrifuged to separate plasma and blood cells. The plasma was aliquoted and stored at −80 °C until further use. The blood cells were gone through Ficoll (GE Healthcare, Cat. No# 17144002) gradient centrifugation after a 1:1 dilution in PBS to isolate peripheral blood mononuclear cells (PBMCs) according to the manufacturer's instructions. PBMCs were cryopreserved in fetal bovine serum (FBS) (Hyclone, Cat. No# SH30084.03) with 10% dimethylsulfoxide (DMSO) (Sigma, Cat. No# D2650-100ML) and stored in liquid nitrogen. Before experiments were performed, aliquots of the plasma samples were heat inactivated at 56 °C for 60 min and then stored at 4 °C.

### Pseudovirus neutralization assay
Pseudovirions were generated by co-transfection of a SARS-CoV-2 Spike expression plasmid and an HIV-luciferase backbone vector into HEK293T cells (ATCC, Cat. No# CRL3216) using the Lipofectamine 2000 transfection reagent (Life Technologies, Cat. No# 11668-019) according to the manufacturer's instructions. Supernatants containing pseudotyped virus were collected 48 h after transfection, filtered and stored at −80 °C. Viral titers were measured based on luciferase activity determined by relative light units (Bright-Lite™ Luciferase Assay system, Vazyme, Cat. No# DD1204). Serially 3-fold diluted mAbs were incubated with SARS-CoV-2 pseudovirus at 37 °C for 1 h. A suspension of HeLa-hACE2 cells at a concentration of 130,000 cells/mL in a medium was directly added to the antibody–virus mixture. After 48 h of incubation, the cells were washed with PBS, followed by the addition of Bright-Lite Luciferase Assay Buffer with Bright-Lite Luciferase Assay Substrate (Vazyme, Cat. No# DD1204). The luminescence was measured with a microplate reader (TECAN i-control 2.0). The mAbs were tested in duplicate wells, and the assay was independently repeated at least twice. The neutralization IC₅₀ was calculated using nonlinear regression (log [inhibitor] vs. normalized response, variable slope) (GraphPad Prism v.8.0).

### Live SARS-CoV-2 neutralization
Live SARS-CoV-2 neutralization was performed using a cytopathic effect (CPE) assay in a biosafety level 3 laboratory. Monoclonal antibodies were tested at 2 μg/mL (Beta) or 4 μg/mL (WT, Delta, Omicron BA.1, and BA.2) start concentration and 11 additional serial 2-fold dilutions. Triplicates of each mAb dilution were mixed with 100 TCID₅₀ of authentic SARS-CoV-2 WT (IME-BJ01 strain, GenBank No. MT291831), Beta (CSTR: 16698.06.NPRC2.062100001), Delta (CSTR.16698.06.NPRC6.CCPM-B-V-049-2105-6), Omicron BA.1 (SARS-CoV-2 strain Omicron CoV/human/CHN_CVRI-01/2022), and BA.2 (SARS-CoV-2 strain Omicron CoV/human/CHN_CVRI-04/2022) strain at 1:1 ratio and incubated for 1 h at 37 °C. The antibody–virus complexes were added to 96-well plates seeded with Vero cells (ATCC, Cat. No# CCL-81, >80% density). After incubation at 37 °C for 4 days, CPEs caused by the infection were visually scored for each well in a blinded fashion. The results were then converted into the percentage of neutralization at a certain mAb concentration, and the mean ± SEM were plotted using nonlinear regression (log [inhibitor] vs. normalized response, variable slope) (GraphPad Prism v.8.0).

### Isolation of RBD-specific single B-cell
PBMCs from vaccinated individuals were thawed and processed to enrich B cells using a pan-B-cell isolation kit (Miltenyi Biotec, Cat. No# 130-101-638) according to the manufacturer's instructions. The enriched B cells were incubated for 60 min at 4 °C in cell staining buffer (1 × PBS, 2% FBS) with biotinylated SARS-CoV-2 spike RBD (Acro Biosystems, Cat. No# SPD-C82E9) after blocking with Human TruStain FcX Fc (Biolegend, Cat. No# 422302) for 10 min at 4 °C. Cells were then stained for 30 min at 4 °C with the following anti-human antibodies (all at a 1:100 dilution): anti-CD19-FITC (Biolegend, Cat. No# 363008, Clone SJ25C1), anti-CD3-Pacific Blue (Biolegend, Cat. No# 300431, Clone UCHT1), anti-CD8-Pacific Blue (Biolegend, Cat. No# 301023, Clone RPA-T8), anti-CD14-Pacific Blue (Biolegend, Cat. No# 325616, Clone HCD14), anti-CD27-PerCP/Cyanine5.5 (Biolegend, Cat. No# 356408, Clone M-T271), streptavidin-APC (Biolegend, Cat. No# 405207) and streptavidin-PE (Biolegend, Cat. No# 405203). The stained cells were washed and resuspended in cell staining buffer and

filtered through a 40-µm cell mesh (BD Biosciences, 352340). Single CD3⁻CD8⁻CD14⁻CD19⁺CD27⁺RBD⁺B cells were gated and sorted into 96-well PCR plates containing 4 µL of lysis buffer (0.5 × PBS, 10 mM DTT, 10 units of RNase Inhibitor (New England Biolabs, Cat. No# M0314L) per well using an ID7000™ software (Sony, v1.1.10) for acquisition and Cell Sorter Software (v.3.1.1) for analysis. The plates were snap-frozen on dry ice and then immediately used for subsequent RNA reverse transcription or stored at −80 °C.

### Antibody amplification, cloning, and expression
Variable genes of Human antibody heavy and light chain were generated by RT-PCR protocol[63]. Briefly, RNA from single B cells was reverse transcribed using High-Capacity cDNA Reverse Transcription Kit (Applied Biosystems, Cat. No# 4368813). The cDNA was amplified by nested PCR for generation of the variable IGH, IGL, and IGK genes. Primers of the second round PCR were modified to contain additional nucleotides overlapping with the expression vectors, and the products were purified and cloned into antibody expression vectors encoding the constant regions of human IgG1 by enzymatic assembly[64]. The IGBLAST program (https://www.ncbi.nlm.nih.gov/igblast/igblast.cgi) was used to analyze germline genes, germline divergence or the degree of somatic hypermutation (SHM), the framework region (FR), and the loop length of CDR3 for each antibody clone.

The paired heavy and light chain expression plasmids were co-transfected into HEK293T cells grown in 12-well plates. After 2 days, the transfected culture supernatants were collected and directly tested for binding and neutralization. MAbs exhibiting neutralizing activity were re-expressed in HEK293F cells (Thermo Fisher Scientific, Cat. No# R79007) and the supernatant was purified using Protein G bead columns (Solarbio, Cat. No# R8300) according to the manufacturer's instructions. Related primers (Supplementary Table 1), VH and VL sequences of anti-SARS-CoV-2 monoclonal antibodies (CB6, C121, COV2-2130, and COVA1-16) were synthesized (Tsingke) and produced in the HEK293F cell system.

### Biolayer interferometry (BLI)
MAb affinity and the competitive binding of mAbs and hACE2 (or between two antibodies) were measured using an Octet RED384 system (FortéBio, Octet data acquisition v12.0) at 30 °C with an orbital shaking speed of 1000 rpm. For the affinity ($K_D$) determination, streptavidin biosensors (Molecular Devices, Cat. No# 18-5019) were loaded with recombinant biotinylated RBD of wild type (Acro Biosystems, Cat. No# SPD-C82E9), Omicron BA.1 (Acro Biosystems, Cat. No# SPN-C82E4), Omicron BA.1 (Acro Biosystems, Cat. No# SPN-C82Eq), and Omicron BA.1 (Acro Biosystems, Cat. No# SPN-C82Ew) (1.0−1.2 nm), respectively. After a baseline step in PBS (Gibco, Cat. No# C10010500BT) for 60 s, the antigen-loaded biosensors were incubated with mAbs for 200 s and then immersed (200 s) into PBS to measure any dissociation of the mAbs from the biosensor surface. Data for which the binding responses were >0.1 nm were aligned, interstep corrected (to the association step) and fitted to a 1:1 binding model using the FortéBio data analysis software, version 12.1.

For the hACE2 competition assay, streptavidin biosensors (Molecular Devices, Cat. No# 18-5019) were immobilized with recombinant biotinylated RBD (Acro Biosystems, Cat. No# SPD-C82E9) (1.0−1.2 nm). The RBD-coated biosensor was incubated with the mAbs (300 nM) for 300 s after a baseline step in PBS buffer for 60 s. After another baseline step in PBS for 60 s, the biosensors were associated with the hACE2 receptor (150 nM) (Sino Biological, Cat. No# 10108) for 300 s. The maximum binding of hACE2 was normalized to a PBS-only control. The percent binding of hACE2 in the presence of the antibody was compared to the maximum binding of hACE2. A reduction in the maximal signal to less than 20% was considered hACE2-blocking.

For the mAbs binding-competition assay, the RBD immobilized sensors (Molecular Devices, Cat. No# 18-5019) were dipped in PBS for

60 s and then incubated with the first antibody (Ab1, 300 nM) for 300 s. After a 60 s baseline step in PBS, the biosensors were immersed into the second antibody (Ab2, 150 nM) for 300 s. Curve fitting was performed using FortéBio Octet data analysis software (FortéBio).

To determine the binding of mAb 8-9D to RBD and RBD mutants, mAb 8-9D was loaded on a protein A biosensor (Octet, Cat. No# MSPP-185012). The biosensor was washed in PBS with 0.05% Tween 20 and 0.1% BSA for 380 s to reach a stable base state. Then the sensors were incubated with RBD-Y505H or N460K for 300 s. The biosensor was then washed with PBS with 0.05% Tween 20 and 0.1% BSA for 300 s to allow the dissociation of the bound RBD mutants. Signals were acquired by the FortéBio Octet data acquisition software and curves were fitted to a 1:1 binding model using the FortéBio data analysis HT v12.1.

### Expression and purification of RBD and Omicron BA.5 spike ectodomain and 8-9D antibody and complex preparation
The heavy chain and light chain of 8-9D were separately cloned into the pCMV vector and transiently transfected into HEK293F cells at a ratio of 1:2 by using PEI. The supernatant containing antibodies was harvested 4 days after transfection. 8-9D was captured by using protein A Sepharose (GE Healthcare) and eluted with 0.1 M glycine at pH 3.2. The elution neutralized to pH 7.6.

To prepare 8-9D Fab fragment, the purified 8-9D was digested by using the protease papain (Sigma, Cat. No# P3125) with an IgG to papain ratio of 66:1 (w/w) for 3 h at 37 °C. The Fc domain and undigested antibodies were removed with protein A Sepharose (GE Healthcare) and the flowthrough was collected.

The extracellular domain of Omicron BA.5 spike (1–1208 amino acids) was cloned into the pCMV vector with residues 817, 892, 899, 942, 986, and 987 mutated to prolines (S6P mutant) and a "GSAS" substitution at residues 682 to 685 with a C-terminal T4 fibritin trimerization motif followed by a flag tag. The RBD (319-541 amino acids) was cloned to the pCDNA3.1 vector after the gp120 signal peptide and followed by a flag tag. The plasmids were transiently transfected to HEK293F cells with polyethylenimine (PEI) (Polysciences, Cat. No# 24765). The recombinant protein was harvested four days after transfection. The spike or RBD was affinity purified from the cell culture by anti-flag antibody beads and eluted by using 0.1 mg/mL 3×Flag peptide in 20 mM HEPES at pH 7.6 and 150 mM NaCl.

For Fab-spike complex assembly, the eluted spike and digested 8-9D Fab were incubated on ice for 20 min and further purified by size-exclusion chromatography using a Superose 6 10/300 column (GE Healthcare) running in buffer with 20 mM HEPES pH 7.6 and 150 mM NaCl. The peak fraction was collected for gradient fixation (GraFix). The concentrated sample was applied to a 10−40% $v/v$ linear glycerol (Sigma, Cat. No# G5516) gradient supplemented with 0.15% glutaraldehyde (Sigma, Cat. No# G5882). After centrifugation at 200,000 × $g$ for 13 h, a peak fraction containing the Fab-spike complex was collected. The sample buffer containing glycerol was exchanged by centrifugation with 20 mM HEPES at pH 7.6 and 150 mM NaCl.

### Cryo-EM sample preparation, data collection, and processing
The sample was concentrated to 1.1 mg/mL. 3 µL of the sample was applied to glow-discharge holey carbon grids (Quantifoil, Cu 200 mesh, R1.2/1.3). The grid was blotted for 3 s in 100% humidity and flash-frozen in liquid ethane by using a Vitt Mark IV (Thermo Fisher).

Micrographs were collected by Titan Krios electron microscope (FEI Company) equipped with a field emission gun operated at 300 kV and a Gatan K3 Summit camera. Images were recorded at a defocus range of −1.5 to −2.0 µm, with the pixel size of 0.97 Å and a total dose of ~50 electrons per Å². 4516 movie stacks were collected by SerialEM v4.0.4 and the frames in each movie stack were aligned, summed, and 2 × binned using Motion Cor2 v1.2.6[65]. The CTF parameters of the micrographs were determined by Gctf v1.18[66].

A total of 1,062,048 particles were picked by using Gautomatch. 2D classification and 3D classification were performed by RELION 3.1[67]. The particles from 2D classification with clear features were selected for 3D classification without symmetry imposed (Supplementary Fig. 2). Two classes from the classifications show better structural details in which spike binding two or three Fabs. The class of spike binding with three Fabs was selected and subjected to cryoSPARC v3.3.2[68] for non-uniform refinement with C3 symmetry imposed and auto-generated mask. The density map was applied with a negative B-factor of 123.5 Å$^2$ by cryoSPARC and the resolution of the reconstructed map was 3.0 Å. However, the density was poor at the interface between 8-9D Fab and RBD. Both of the particles from 2Fab and 3Fab classes can be used for cryoSPARC local refinement. A soft mask focused on the Fab-RBD subunit generated by RELION (v3.1) was applied to refined particles for local refinement with C1 symmetry imposed. The resolution of the interface was improved to 3.32 Å with a negative B-factor of 142.3 Å$^2$ applied. The local resolutions of the cryo-EM maps were calculated by using cryoSPARC with a threshold of 0.143. For model building, the unsharpened maps were post processed by deepEMhancer[69]. The atomic model was built and adjusted by using COOT (v0.9.6)[70]. The model was refined by using the PHENIX cryo-EM Real-space Refinement tool (v1.19)[71]. Due to the flexibility of the constant region of the Fab fragment, only the variable region was modeled in the final atomic coordinate. The interactions between RBD and mAb 8-9D were analyzed by PISA[72] and CCP4i2 v.1.0.2[72]. All structural figures were generated by ChimeraX v.1.3[73]. Details of the data collection and processing are shown in Supplementary Table 5.

### Structural modeling of mutations on BA.5 RBD

We mutated target residues and chose possible orientations from the COOT rotamer library. The mutated residue and residues in distance less than 4 Å were selected for energy minimization by using Foldit Standalone[74] with both main chain and side chain adjustable.

### Generation of modified mRNA

The linearized mRNAs of 8-9D were produced in vitro using T7 polymerase-mediated transcription from linearized DNA templates, and the UTPs were completely replaced with 1-methylpseudoUTPs. The incorporated 5′-cap1 modifications were added by the ScriptCap™ Cap 1 Capping System (CELLSCRIPT, Cat. No# C-SCCS1710) for better mRNA stability and translation efficiency. mRNA templates were constructed as such: for the 8-9D-H (8-9D heavy chain), a Flag tag was inserted before the stop codon; for 8-9D-L (8-9D light chain), a V5 tag sequence was set at the C-terminus. The CircRNA8-9D precursor was synthesized through in vitro transcription (IVT) using a permuted intron-exon system engineered as previously described by Wessel-hoeft et al. Following IVT, the reaction was treated with DNase I (Takara) for 20 min according to the manufacturer's instructions. Subsequently, the reaction was heated to 70 °C for 5 min and immediately placed on ice for 3 min. GTP was then added at a final concentration of 2 mM along with a buffer containing magnesium (50 mM Tris-HCl, 10 mM MgCl$_2$, 1 mM DTT, pH 7.5) for RNA circularization at 55 °C for 15 min. To enrich circRNA, high-performance liquid chromatography (HPLC) was performed using a Sepax Technologies' size-exclusion column measuring at dimensions of 7.8 × 300 mm with particle size of 5 μm and pore size of 1000 Å on an Agilent1100 Series HPLC instrument (Agilent). RNA samples were run in RNase-free PBS buffer (150 mM sodium phosphate, pH 7.0) at a flow rate of 0.6 mL/minute and detected via UV absorbance at 260 nm. CircRNA8-9D fraction was collected, purified, and precipitated using 5 M ammonium acetate before being resuspended in water.

### SDS-PAGE and western blotting

In vitro expression of the 8-9D linearized mRNA in HEK293T cells transfected using TransIT®-mRNA Reagent (Mirus Bio, Cat. No# MIR-2250) was confirmed by SDS-PAGE and western blotting. Briefly, HEK293T cells were plated in 6-well plates 24 h prior to transfection, and the medium was changed to Opti-MEM (Gibco, Cat. No# 31985062) 2 h before transfection. One microgram of 8-9D-H (8-9D heavy chain) and/or 8-9D-L (8-9D light chain) mRNA complexed with TransIT®-mRNA Reagent were used to transfect cells. 72 h later, the supernatant was mixed with 5 × SDS loading buffer denatured at 100 °C for 10 min for the SDS-PAGE or Western blot analysis. Antibodies used in the Western blot analysis were anti-Flag tag rabbit mAb (for 8-9D-H mRNA, Proteintech, Cat. No# HRP-66008), and an anti-V5 tag mouse mAb (for 8-9D-L mRNA, GenScript, Cat. No# A01733).

### Synthesis of the ionizable lipid and targeting lipid

Oleic acid (28.2 g, 100 mmol) and thionyl chloride (17.8 g, 150 mmol) and 3 drops of N,N-dimethylformamide were dissolved in 150 mL of toluene. The solution was heated under stirring at 60 °C for 4 h, after which the toluene and thionyl chloride were evaporated under vacuum to give the oleoyl chloride (28.6 g, 95.1%) which was used without further purification.

Oleoyl chloride (6.0 g, 20.0 mmol) and triethanolamine (1.50 g, 10.0 mmol) were dissolved in dichloromethane (DCM), into which triethylamine (2.20 g, 22.0 mmol) was added dropwise within 1 h under stirring at 0 °C. After stirring at room temperature overnight, the reaction was filtered and the solution was diluted with DCM washed with water. The organic layer was dried over anhydrous Na$_2$SO$_4$ and then evaporated to afford the crude product, which was isolated by flash column chromatography using DCM/methanol (9:1) to give the ionizable lipid as brown liquid (5.30 g, 78.2%). The $^1$H NMR spectrum of the ionizable lipid is shown in Supplementary Fig. 15a. The $^{13}$C NMR spectrum of the ionizable lipid is shown in Supplementary Fig. 15b. ESI-TOF MS: $m/z$ calcd for [M + H]$^+$ C$_{42}$H$_{80}$NO$_5$, 678.6031, found 678.6013 (Supplementary Fig. 15c).

Synthesis of the cationic targeting lipid: Ionizable lipid (678 mg, 1.00 mmol) and methyl iodide (282 mg, 2.00 mmol) was dissolved in 10 mL of acetonitrile and the solution was heated at 70 °C for 24 h, after that the solvent and methyl iodide were evaporated to give the final product targeting lipid (803 mg, 98.0%). The $^1$H NMR spectrum of the ionizable lipid is shown in Supplementary Fig. 15d. The $^{13}$C NMR spectrum of the ionizable lipid is shown in Supplementary Fig. 15e. ESI-TOF MS: $m/z$ calcd for [M + H]$^+$ C$_{43}$H$_{82}$NO$_5$, 692.6188, found 692.6160 (Supplementary Fig. 15f).

### Lipid nanoparticle preparation

The mRNA was dissolved in a 100 mM citrate buffer (pH 5.0). The Liver-LNPs lipid ethanol solution contains ionizable lipid, DSPC, cholesterol, and PEG-lipid with a molar ratio of 49.1%/9.4%/40.0%/1.5%. The Lung-LNPs lipid ethanol solution contains ionizable lipid, DOPE, cholesterol, PEG-lipid, and targeting lipid with a molar ratio of 24.5%/4.7%/20.0%/0.8%/50%. The mRNA-containing buffer and the ethanol solution were mixed rapidly at an aqueous-to-ethanol ratio of 3/1 by volume (3/1, aq./ethanol, $v/v$) with a weight ratio of 40/1 (total lipids/mRNA). The obtained LNPs@mRNA was dialyzed against 1 × PBS for 2 h for further use.

### LNPs characterization

For the transmission electron microscope test, the prepared LNPs@mRNA was dialyzed against Milli-Q water for 2 h. Then the dialyzed LNPs@mRNA was diluted 100 times using Milli-Q water and 10 μL of the sample was dropped onto a carbon-coated copper grid. Samples were dried under a vacuum to remove water. Tests were carried out on a Hitachi HT-7700. For dynamic light scattering analysis and zeta potential analysis, dialyzed LNPs@mRNA was diluted 10 times using Milli-Q water, and tests were carried out on Zetasizer Nano-ZS90 (Malvern Instruments, UK).

## Luciferase or eGFP mRNA delivery

C57BL/6 mice with weights of 20–22 g were i.v. injected with Liver-LNPs@mRNA$^{Luci}$ or Liver-LNPs@mRNA$^{Luci}$ (5 μg/dose). $_D$-Luciferin (Yeasen, Cat. No# 40901ES01, 150 mg kg$^{-1}$) was administered to mice at the stated time points, and imaging tests were performed using an IVIS Lumina system (Perkin Elmer). The blood was collected at 6 h post injection for kidney and liver injury analysis. Similar procedures were applied for the eGFP mRNA delivery, and collected lung or liver samples were subjected to confocal laser scanning microscope analysis. For the analysis of cell types transfected in the lungs, 2 or 6 h after intravenously injecting Lung-LNPs@mRNA$^{eGFP}$, mice were sacrificed. Lungs were harvested and ground in the FACS buffer. The obtained solution was filtered using a 70 μm cell strainer and processed using red blood lysis buffer for 5 min. The samples were then centrifuged at $700 \times g$ and subsequently incubated with a mixture, including Fixable Viability Dye eFluor™ 780, antibodies against epithelial (EpCAM, Biolegend, Cat. No# 118214), immune (CD45, Biolegend, Cat. No# 157605, Clone QA17A26), and endothelial (CD31, Biolegend, Cat. No# 102409, Clone 390) cell markers at 4 °C for 30 min. The stained cells were washed twice with cold FACS buffer and resuspended in a 4% paraformaldehyde fix solution. Flow cytometry (BD Biosciences, USA) was used to determine the internalization of LNPs@mRNA$^{eGFP}$ in various cell subtypes. The cells were first gated for live single cells, followed by gating strategies for different cell types. Finally, the proportion of eGFP-positive cells in each cell subtype was counted and statistically graphed ($n = 3$).

## ELISA

The 8-9D levels in the sera, BALF, livers, and lungs were measured by ELISA. Briefly, 96-well microtiter plates were coated overnight with RBD protein diluted in 0.1 M carbonate-bicarbonate buffer (2 μg/mL). After overnight coating, wells were washed and blocked with 2% non-fat milk in phosphate-buffered saline (PBS) for 2 h. Next, serially diluted samples containing the 8-9D (sera, BALF, liver lysis buffer, and lung lysis buffer) were added to the wells and incubated for 2 h. The bound 8-9D were detected with horseradish peroxidase (HRP)-linked anti-human IgG (Invitrogen, Cat. No# A18817). The purified 8-9D antibody was set for the standard curve, served for the quantitative calculation.

## Indicators of kidney and liver injury

For the liver and renal function evaluation, sera were collected at 6 h post injection, levels of alanine aminotransferase (ALT), aspartate aminotransferase (AST), uric acid (UA), blood urea nitrogen (BUN), and creatinine (CREA) were measured using a clinical chemistry analyzer - Idexx VetTest 8008 (Idexx Laboratories, Westbrook, ME, USA).

## Erythrocyte hemolysis

The hemolytic effect of LNP was evaluated according to previous reports with some modifications[49,51]. Blood samples were obtained from healthy donors. The erythrocytes were separated by centrifugation at $150 \times g$ for 8 min and washed three times with four volumes of PBS. thereafter lipid nanoparticles were immediately dissolved in the erythrocyte suspension (up to 20% of nanoparticles in erythrocyte suspension; $v/v$). Incubations were run for 1 h at 37 °C with gentle tumbling of the test tubes, Samples were then diluted in dichloromethane and centrifuged for 8 min at $150 \times g$ to avoid the interference of lipid nanoparticles in absorbance. The absorbance of the supernatant was measured at 415 nm to determine the percentage of cells undergoing hemolysis. Hemolysis induced with distilled water was used as a positive control.

## Inflammatory response to Liver-LNPs@mRNA$^{8-9D}$ or Liver-LNPs@mRNA$^{8-9D}$

For the cytokine measurements, serum was collected at 6 h after i.v. injection of the Liver-LNPs@mRNA$^{8-9D}$ or Liver-LNPs@mRNA$^{8-9D}$. A panel of 13 cytokines, including IL-1α, IL-1β, IL-6, IL-10, IL-12p70, IL-17A, IL-23, IL-27, monocyte chemoattractant protein 1 (MCP-1), interferon beta (IFN-β), IFN-γ, tumor necrosis factor alpha (TNF-α), and granulocyte-macrophage colony-stimulating factor (GM-CSF), was analyzed by a multiplex assay according to the manufacturer's instructions (Biolegend, Cat. No# 740446).

For the pulmonary immune cell subset analysis, lungs were collected at 72 h after injection of the Liver-LNPs@mRNA$^{8-9D}$ or Liver-LNPs@mRNA$^{8-9D}$. The lungs were dissected and homogenized. Single-cell suspensions were stained with a LIVE/DEAD Ghost Dye™ UV 450 (Tonbo Biosciences, Cat. No# 13-0868-T500, 1:1000) to exclude dead cells from analysis, and then incubated with the antibody panel of CD45-AF700 (Invitrogen, Cat. No# 56-0451-82, Clone 30-F11), CD11c-AF488 (Biolegend, Cat. No# 117313, Clone N418), CD11b-Pacific Blue (Biolegend, Cat. No# 101223, Clone M1/70), CD24-PE/Cyanine7 (Elabscience, Cat. No# E-AB-F1179UH, CloneM1/69), CD64-PE (Elabscience, Cat. No# E-AB-F1186UD, X54-5/7.1), and I-A/I-E-AF647 (Biolegend, Cat. No# 107618, Clone M5/114.15.2) for 30 min at room temperature. After washing steps, the cells were analyzed on the Sony ID7000™ Spectral Cell Analyzer equipped with 5 lasers (355 nm, 405 nm, 488 nm, 561 nm, 637 nm). The unmixing of samples was done using the Sony ID7000™ software (v1.1.10), then unmixed FCS files were exported and data were analyzed for different pulmonary immune subsets using the FlowJo software V10 (Treestar, Woodburn, OR, USA).

## SARS-CoV-2 challenge

Studies with authentic SARS-CoV-2 Beta strain and Omicron BA.2 were performed in biosafety level 3 laboratories. Female 4 to 6-week-old K18-hACE2 transgenic mice were randomly allocated to groups ($n = 5$). For the prophylactic study, the mice were intravenously administered one dose of 5 μg of Liver-LNPs@mRNA$^{8-9D}$ or Lung-LNPs@mRNA$^{8-9D}$, or PBS alone as a control. After 24 h, the mice were anesthetized and intranasally inoculated with $2 \times 10^4$ TCID$_{50}$ of authentic SARS-CoV-2 Beta variant or Omicron BA.2 variant. The mice were euthanized 4 days post infection to harvest lung tissues and trachea tissues for virological assessment or histopathology evaluation. SARS-CoV-2 E gene sgRNA was quantified by quantitative reverse transcription-PCR (qRT–PCR) assays (sgRNA-F: CGATCTCTTGTAGATCTGTTCTC, sgRNA-F:ATATTG CAGCAGTACGCACACA). Standard curves were generated using serial dilutions of a cDNA plasmid of known concentration and used to calculate the level of sgRNA in copies per gram. For the post-exposure treatment, the mice were first anesthetized and intranasally inoculated with $2 \times 10^4$ TCID$_{50}$ of authentic SARS-CoV-2 Beta variant or Omicron BA.2 variant, and after 24 h, the mice were intravenously administered one dose of 5 μg of Liver-LNPs@mRNA$^{8-9D}$ or Lung-LNPs@mRNA$^{8-9D}$, or PBS alone as a control. The mice were euthanized 4 days post infection to harvest lung tissues and trachea tissues for virological assessment or histopathology evaluation.

## Immunofluorescence staining and microscopy

HEK293T cells were plated in 6-well plates 24 h prior to transfection, and the medium was changed to Opti-MEM (Gibco, Cat. No# 31985062) 2 h before transfection. One microgram of 8-9D-H (8-9D heavy chain) and 8-9D-L (8-9D light chain) mRNA complexed with TransIT®-mRNA Reagent were used to transfect cells. 72 h later, the cells were subjected to immunofluorescence staining. Antibodies used were anti-Flag tag rabbit mAb (for 8-9D-H mRNA, Proteintech, Cat. No# 80010-1-RR, Clone 4K14), an anti-V5 tag mouse mAb (for 8-9D-L mRNA, Invitrogen, Cat. No# 37-7500, Clone 2F11F7), an Alexa Fluor 488 labeled anti-mouse IgG (H + L) cross-adsorbed secondary antibody (Invitrogen, Cat. No# A-11001), and an Alexa Fluor 594 labeled anti-rabbit IgG (H + L) secondary antibody (Invitrogen, Cat. No# A-21207). The nuclei were stained with DAPI (Beyotime, Cat. No# C1005). Images were captured using a Zeiss LSM 880 meta confocal microscope.

## Statistical analysis

Results are presented as mean ± SEM or mean ± SD. Ordinary one-way ANOVA with Tukey's test was for multiple group comparisons, and a description of the statistical analysis is provided in the figure legends. All statistical analyses were performed with Prism 8.0.1 software (GraphPad).

## Reporting summary

Further information on research design is available in the Nature Portfolio Reporting Summary linked to this article.

## Data availability

The authors declare that all data supporting the results in this study are available in the paper and Supplementary Materials. The data generated in this study are provided in the Supplementary Information and Source data file. The atomic coordinates and EM maps have been deposited into the Protein Data Bank (http://www.pdb.org) and the EM Data Bank, respectively: The whole complex of spike binding with three 8-9D Fabs (PDB: 8J1V, EMD: 35934), local refined 8-9D Fab-RBD complex (PDB: 8J1T, EMD: 35932). Sequences of the monoclonal antibody 8-9D characterized here has been deposited at GenBank (accession number: OQ868201 and OQ868202). Source data are provided with this paper.

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

## Acknowledgements

This work was funded by grants from the National Key Research and Development Plan of China (2021YFC2300200, 2021YFC12302405, 2022YFC2303200 and 2022YFC2303400) to G.C. The National Natural Science Foundation of China (32188101, 81961160737 and 31825001) to G.C. The Ministry of Science and Technology of China (2021YFA1300204), the National Natural Science Foundation of China (31925023, 21827810 and 31861143027) to Y.X. The National Natural Science Foundation of China (82271872 and 32100755) to W.T. The National Key Research and Development Plan of China (2022YFC2303400) to Y.L. Vanke Special Fund for Public Health and Health Discipline Development, Tsinghua University (2022Z82WKJ005, 2022Z82WKJ013) to G.Y. National Natural Science Foundation of Guangdong, China (2021A1515010478) to J.W. Tsinghua University Spring Breeze Fund (2020Z99CFG017, 2021Z99CFZ007), Shenzhen Science and Technology Project (JSGG20191129144225464) and Shenzhen San-Ming Project for Prevention and Research on Vector-borne Diseases (SZSM202211023) to G.C. Innovation Team Project of Yunnan Science and Technology Department (202105AE160020) and the Yunnan Cheng Gong expert workstation (202005AF150034) to G.C. This work was financially supported by XPLORER PRIZE from Tencent Foundation.

## Author contributions

W.T., K.Y., G.Y., and G.C. conceived and designed the experiments. W.T., K.Y., Y.L., R.L., S.F., B.C., X.Z., S.Q., H.S., Z.L., E.M., W.W., C.T., T.L., J.L., J.W., Y.C., and M.T. performed the experiments. W.T., K.Y., Y.L., X.Z., R.L., S.F., M.T., Y.X., G.Y., and G.C. analyzed the data. W.T., K.Y., Y.L., R.L., Y.X., G.Y., and G.C. wrote the manuscript. All authors have given approval to the final version of the manuscript.

## Competing interests

G.Y., G.C., W.T., K.Y., and Y.L. have applied for the patent (202310810894X) related to the lung-selective LNP formulation and its preparation. All other authors declare that they have no competing interests.
