## [Peer Review File · Nature Communications]

A lung-selective delivery of mRNA encoding broadly neutralizing antibody against SARS-CoV-2 infectionReviewers' Comments:

Reviewer #1:

Remarks to the Author:

In the manuscript entitled, "A lung selective delivery of mRNA encoding broadly neutralizing antibody against SARS-CoV-2 infection", the authors discuss the discovery of antibody 8-9D and the use of a lung targeted LNP for delivery of mRNA encoded 8-9D and protection from SARS-CoV-2.

Overall this is a well-executed set of experiments but there are a number of issues that affect the translational potential of this approach and there are technical issues that need to be addressed.

Major issues.

One major issue is the use of IV delivery and the very short half life of the antibody in the BALF and the lung.

1. First, the authors need to define the translational use case for this approach in the introduction. When would it make sense to give people this drug IV, as prophylaxis, meaning they would have to be in a medical setting?
2. Second, the half life in the lung was very short. It peaked on day 2 but was gone by day 4 or 5. If you were trying to protect people from infection, they would have to receive this very often to keep the concentration high enough to possibly prevent infection. This would likely not be feasible. This issue would have to be addressed or the approach would never be used as presented. Or this would have to be used as a treatment, post-infection.

Technical issues.

1. Please identify specifically the cell types being transfected. In most cases, when a charged LNP is being used, they preferentially make it to the lung, but they tend to transfect endothelial cells not epithelial cells. Can you identify which cell types are being transfected? Is this an issue for this approach, meaning is it restricting the amount of antibody making it to the lung epithelium, where the infection takes place?
2. Can you address the short half-life of the Ab in the BALF and lung? Most Ab (protein) approaches now have very long half lives.
3. Can you demonstrate that this approach is effective post-infection, as it is unlikely to be used, as presented, as prophylaxis?
4. Can you clarify the timing for the evaluation of both cytokines in the blood, whether cytokines in the lung were measured, and for the evaluation of cell infiltrates? It was not clear what times were examined and why? The timing is critical. Please present a rational and add time points to clarify what is happening. Highly charged LNPs often have toxicity effects. Also, how does this change with dose?

Reviewer #2:

Remarks to the Author:

The authors present the structure of a broadly neutralizing antibody, isolated from a vaccinated patient. The structure is of adequate quality and appears to be of sufficient resolution to interpret the binding contacts between the nAb and RBD (however, please note questions to address below). The authors use re-formulated lipid nanoparticles to deliver mRNA encoding this antibody directly to lung tissues, which enhances the protective efficacy of the nAb against Beta and BA.2 variants. Overall, the results and interpretations are clear.

One weakness of the manuscript is the potential limited novelty and impact. Major points to address

are:

- 1) The authors state that "Lung-selective mRNA encoding broadly neutralizing antibodies can directly induce protection against SARS-CoV-2 in the lungs" (lines 231-232). They do not discuss how they have improved upon these previous publications (nor do they cite them for reference).
- 2) The authors show the prophylactic efficacy of the lung-targeted mRNA is excellent; however, the antibody titer drops to baseline within 8 days of vaccination, limiting the use of the mAb to a very short window. An experiment to show neutralizing activity of the serum from immunized animals for each timepoint in Figure 4g and 4h would provide some indication of how long the effect of vaccination might last. Additional discussion would also be beneficial to address the applicability to post-exposure treatment (the most likely scenario).

Comments related to the structure:

- 1) Overall, general statistics look ok, but it is difficult to tell if the residues are well placed without the map/model (the link was not functional to retrieve them). In particular, the model with the whole spike has a significant number of flags. In contrast, the local refinement looks better (as expected). Did the authors use the model from local refinement to fit the entire spike structure globally? This might fix the issues unless there is a map quality issue or density missing. It could be that the map/model for the Spike structure is ok. However, based on the validation report, I would suggest that they do not model the Fabs on the global map but only the map from local refinement.
- 2) It is not clear to me why the number of particles for local refinement is 1.6M, while for the global model is 200k. This could use some explanation.
- 3) Page 37 of the supplement (global structure primary map): I am unfamiliar with this plot, but it shows some strange map behavior. Has the map been density modified? If so, using modified maps as the main map in EMDB is incorrect. However, it might be some new way EMDB shows maps, so this needs to be checked manually.

Additional minor comments:

- 1) The manuscript would benefit from more careful editing. There are several typos and confusing sentences throughout. I have highlighted some of these in the attached annotated file, but the authors should carefully edit to ensure clarity.
- 2) Figure 4 would benefit from additional labels and in some cases, smaller fonts. A study design panel would also be helpful. Some panels could be moved to supplementary figures to accommodate the study design.
- 3) Mice received 5ug doses of LNP-mRNA – how does this compare with the current 50-100ug dose of mRNA COVID vaccines?

Reviewer #3:

Remarks to the Author:

This paper demonstrates that LNPs with lung-selective mRNA delivery ability produce neutralizing antibodies in the lungs against SARS-CoV-2 infection. Although there have already been papers on LNPs with lung-selective mRNA delivery capability, this paper is valuable in that it uses Lung-LNPs with novel substances and lipid compositions, and demonstrates their therapeutic efficacy against SARS-CoV-2 infection in animals and the safety of the carriers themselves. If the questions in the following points are resolved, the paper may be published in Nature Communication.

Specific points:

1. As shown in Figure 3h, the positive charge of Lung-LNPs is very high (25mV). In the past, cationic liposomes with such a high positive charge had problems with hemolytic properties due to interaction with negatively charged erythrocytes. Therefore, the authors should add the data and discussion by citing previous reports on the interaction with erythrocytes.

2. The authors should add data and discussion regarding the lipid composition of Lung-LNPs determined in this study for future development of Lung-LNPs research.

Reply to Reviewer 1:

Reply to the first comment made by Reviewer 1: “In the manuscript entitled, “A lung selective delivery of mRNA encoding broadly neutralizing antibody against SARS-CoV-2 infection”, the authors discuss the discovery of antibody 8-9D and the use of a lung-targeted LNP for delivery of mRNA encoded 8-9D and protection from SARS-CoV-2. Overall, this is a well-executed set of experiments but there are a number of issues that affect the translational potential of this approach and there are technical issues that need to be addressed.”

Response:

We thank the positive comments from reviewer 1.

Reply to the second comment made by Reviewer 1: “First, the authors need to define the translational use case for this approach in the introduction. When would it make sense to give people this drug IV, as prophylaxis, meaning they would have to be in a medical setting?”

Response:

Thanks for the suggestion, we have added relevant descriptions in Introduction to define the translational use case.

Reply to the third comment made by Reviewer 1: “Second, the half-life in the lung was very short. It peaked on day 2 but was gone by day 4 or 5. If you were trying to protect people from infection, they would have to receive this very often to keep the concentration high enough to possibly prevent infection. This would likely not be feasible. This issue would have to be addressed or the approach would never be used as presented. Or this would have to be used as a treatment, post-infection.”

Response:

This lung-targeted delivery of antibody-encoded mRNA technology is suitable for both prophylaxis and post-therapy. In comparison with current antigen-based mRNA vaccines which suffer from the problems of immune escaping and the long

period to produce enough antibodies, our strategy produces immediate and potent protection against various strains of the virus in the lungs. In the case of treatment after infection, the rapid antibody responses in the lungs eliminate the virus, and the sustaining antibodies further avoid the virus invasion. Moreover, the circular mRNA is much more stable than linear mRNA, which can also elongate the half-life of the mRNA vaccine apparently based on our previous research (*Theranostics* 2022, 12, 6422-6436). To improve the antibody half-life in the lungs, we conducted this experiment and successfully extended the half-life of the antibody to nearly two weeks (**Extended Data Fig. 9**). The relevant results have confirmed the feasibility of further clinical translation of this study. In addition, to further demonstrate the effectiveness of targeted antibody delivery for post infection applications, we included protective studies involving administration after infection (**Fig. 5e**). The results revealed that administration of Lung-LNPs@mRNA^{8-9D} 24 h after being infected with the coronavirus could also provide substantial protection, resulting in a protection rate approaching 100% (**Fig. 5f-h, Extended Data Fig. 14d-f**).

Reply to the fourth comment made by Reviewer 1: “Please identify specifically the cell types being transfected. In most cases, when a charged LNP is being used, they preferentially make it to the lung, but they tend to transfect endothelial cells not epithelial cells. Can you identify which cell types are being transfected? Is this an issue for this approach, meaning is it restricting the amount of antibody making it to the lung epithelium, where the infection takes place?”

Response:

In the revised manuscript, we quantitatively analyzed the transfected percentage of different cells within lungs by flow cytometry using Lung-LNPs@mRNA^{eGFP} (**Extended Data Fig. 10**). After the injection of Lung-LNPs@mRNA^{eGFP}, the analysis of eGFP-expressing cell types in the lungs using flow cytometry revealed a transient significant increase in endothelial cell expression at 2 h post injection, and by the 6 h mark, the expression levels across various cell types tended to reach a balance among

epithelial cells, endothelial cells and lymphocytes. Moreover, the mRNA-encoding antibody would be secreted after translation, which was proved by the alveolar lavage experiment (**Figure 4**). Even though the lung-targeting delivery system is localized to a specific lung cell subset, the virus can be neutralized timely. Therefore, our lung-targeting delivery system will not restrict the amount of antibodies making it to the lung epithelium, where the infection takes place.

Reply to the fifth comment made by Reviewer 1: “Can you demonstrate that this approach is effective post-infection, as it is unlikely to be used, as presented, as prophylaxis?”

Response:

We conducted experiments to demonstrate that our strategy is also effective in protecting the mice post-infection (**Fig. 5e-h, Extended Data Fig. 14d-f**). The results revealed that administration of Lung-LNPs@mRNA^{8-9D} 24 h after being infected with the coronavirus could also provide substantial protection, resulting in a protection rate approaching 100%. This new technology resolved the immune deficiency period after vaccination because the current antigen-based mRNA vaccines take several days to produce enough antibodies to combat virus, while it takes merely 2 hours to produce broadly neutralizing antibodies using this strategy, showing unparalleled advantages.

Reply to the sixth comment made by Reviewer 1: “Can you clarify the timing for the evaluation of both cytokines in the blood, whether cytokines in the lung were measured, and for the evaluation of cell infiltrates? It was not clear what times were examined and why? The timing is critical. Please present a rationale and add time points to clarify what is happening. Highly charged LNPs often have toxicity effects. Also, how does this change with dose?”

Response:

We agree with the reviewer's mention of the impact of sample collection timing on the conclusions. Immediate samples can reflect relevant safety concerns. Therefore, we presented the results of cytokine detection at 6 hours post-injection (**Extended**

Data Fig. 11) and the extent of pulmonary inflammatory cell infiltration at the corresponding time point (**Extended Data Fig. 12**). Additionally, we included measurements of three key cytokines (IL-6, TNF- α and IL-1 β) levels at 1 day, 2 days, 4 days, and 8 days post injection (**Fig. 4h-j**) to demonstrate the safety of the formulation in this study. Moreover, concentration-dependent hemolysis assay was conducted to evaluate the in vivo safety of lung-targeting LNPs (**Extended Data Fig. 13**). The results demonstrated that hemolysis was negligible even at higher concentrations, further substantiating the safety concerns.

Reply to Reviewer 2:

Reply to the first comment made by Reviewer 2: “The authors present the structure of a broadly neutralizing antibody, isolated from a vaccinated patient. The structure is of adequate quality and appears to be of sufficient resolution to interpret the binding contacts between the nAb and RBD (however, please note questions to address below). The authors use re-formulated lipid nanoparticles to deliver mRNA encoding this antibody directly to lung tissues, which enhances the protective efficacy of the nAb against Beta and BA.2 variants. Overall, the results and interpretations are clear.”

Response:

We thank the positive comments from reviewer 2.

Reply to the second comment made by Reviewer 2: “One weakness of the manuscript is the potential limited novelty and impact.”

Response:

As this work is at the interface of nanotechnology, chemistry, immunology, and epidemiology, it unifies multiple disciplines to provide an exciting and new design that can be used to explore the antiviral therapeutics. Given the themes of broadly

neutralizing antibody, lung-targeting delivery system, and mRNA therapy, we believe this work will appeal to the broad readership of *Nature Communications* and be of interest to immunologists, material scientists, biologists and students at all levels.

This work exhibits unparalleled advantages and novelty listed as follows:

1) Given the low feasibility of targeted delivery of antibodies into the lungs by intravenous administration and the short half-life period of antibodies in the lungs by aerosolized immunization, our work provided a novel and feasible strategy for preventing and treating infectious diseases, such as SARS-CoV-2. Lung-targeting LNPs were fabricated for the delivery of mRNA-encoding broadly neutralizing antibody against SARS-CoV-2, generating high-titer antibody *in situ* in lung and neutralizing the SARS-CoV-2 variants. *In vivo* studies demonstrated our strategy effectively protected K18-hACE2 transgenic mice from the challenge with the Beta or Omicron BA.2 variant of SARS-CoV-2.

2) We firstly identified a human monoclonal antibody from clinical patients, 8-9D, with broad neutralizing potency against SARS-CoV-2 variants. The neutralization mechanism of this antibody was also explained by the structural characteristics of 8-9D Fabs in complex with the Omicron BA.5 spike.

3) Our methods resolve the immune deficiency period after vaccination of the current SARS-CoV-2 vaccines because the current antigen-based vaccine takes more than seven days to produce antibodies. Our work provides rapid and potent antiviral protection as prophylaxis or treatment on the second day after administration, and antibodies can be monitored after two hours of administration. Furthermore, the current antibody administration method is rather expensive, which is not feasible in a pandemic, our strategy is much cheaper and more effective.

This work provides a robust and feasible strategy to extend its biomedical applications in the treatment of various diseases. For example, broadly neutralizing antibodies show promising potentials in the treatment of HIV and other severe diseases (*Science* 2019, 366, 6470; *Nature* 2021, 597, 97–102), however their clinical use is limited by the short half-life of antibodies. mRNA-based therapeutics have been confirmed as powerful and versatile tools to overcome the obstacles to combat these

diseases. We do believe our design can imply a new avenue for treating these thorny diseases.

Reply to the third comment made by Reviewer 2: “The authors state that “Lung-selective mRNA encoding broadly neutralizing antibodies can directly induce protection against SARS-CoV-2 in the lungs” (lines 231-232). They do not discuss how they have improved upon these previous publications (nor do they cite them for reference).”

Response:

The corresponding discussion is included in the Discussion section (lines 400–421), and the corresponding references were cited in the revised manuscript.

Reply to the fourth comment made by Reviewer 2: “The authors show the prophylactic efficacy of the lung-targeted mRNA is excellent; however, the antibody titer drops to baseline within 8 days of vaccination, limiting the use of the mAb to a very short window. An experiment to show neutralizing activity of the serum from immunized animals for each timepoint in Figure 4g and 4h would provide some indication of how long the effect of vaccination might last. Additional discussion would also be beneficial to address the applicability to post-exposure treatment (the most likely scenario).”

Response:

The corresponding discussion has been added. Moreover, we conducted additional experiments to demonstrate that our strategy is also effective in protecting the mice post-infection (**Fig. 5e-h, Extended Data Fig. 14d-f**). Moreover, the circular mRNA is much more stable than linear mRNA, which can also elongate the half-life of the mRNA vaccine apparently based on our previous research (*Theranostics* 2022, 12, 6422-6436). To improve the antibody half-life in the lungs, we conducted this experiment and successfully extended the half-life of the antibody to nearly two weeks (**Extended Data Fig. 9**). This new technology resolves the immune deficiency

period after vaccination because the current antigen-based mRNA vaccines take several days to produce enough antibodies to combat virus, while it takes merely 2 hours to produce broadly neutralizing antibodies using this strategy, showing unparalleled advantages.

Reply to the fifth comment made by Reviewer 2: “Overall, general statistics look ok, but it is difficult to tell if the residues are well placed without the map/model (the link was not functional to retrieve them). In particular, the model with the whole spike has a significant number of flags. In contrast, the local refinement looks better (as expected). Did the authors use the model from local refinement to fit the entire spike structure globally? This might fix the issues unless there is a map quality issue or density missing. It could be that the map/model for the Spike structure is ok. However, based on the validation report, I would suggest that they do not model the Fabs on the global map but only the map from local refinement.”

Response:

We thank for the recognition of the work about structure analysis. We provided the map/model link (doi.org/10.5281/zenodo.7988537), and we did use the atomic model from local refinement to fit the density from global map for Spike-3Fab model, since the Fab density of global map is smear.

Reply to the sixth comment made by Reviewer 2: “It is not clear to me why the number of particles for local refinement is 1.6M, while for the global model is 200k. This could use some explanation.”

Response:

As we mentioned in Method "Both of the particles from 2Fab and 3Fab classes can be used for cryoSPARC local refinement" and Supplementary figure 2 "The workflow of cryo-EM data processing", in local refinement, we regard each RBD-Fab subunit as one particle. Thus, the particles used in local refinement are from 3Fab classes (296568*3) and 2Fab class (360152*2).

Reply to the seventh comment made by Reviewer 2: “Page 37 of the supplement (global structure primary map): I am unfamiliar with this plot, but it shows some strange map behavior. Has the map been density modified? If so, using modified maps as the main map in EMDB is incorrect. However, it might be some new way EMDB shows maps, so this needs to be checked manually.”

Response:

All of the maps are not density modified, we used deepEMhancer to post process unsharpened maps, which caused the marginal density at low counter level to look different from raw maps. We apologize for missing the description about map postprocessing in Method, and we have added the sentence in the manuscript: “For model building, the unsharpened maps were post processed by deepEMhancer” before “The atomic model was built and adjusted by using COOT”.

Reply to the eighth comment made by Reviewer 2: “The manuscript would benefit from more careful editing. There are several typos and confusing sentences throughout. I have highlighted some of these in the attached annotated file, but the authors should carefully edit to ensure clarity.”

Response:

We carefully revised and polished the manuscript.

Reply to the ninth comment made by Reviewer 2: “Figure 4 would benefit from additional labels and in some cases, smaller fonts. A study design panel would also be helpful. Some panels could be moved to supplementary figures to accommodate the study design.”

Response:

The corresponding corrections were done. We separated the Fig.4 into current Fig.4 and Fig.5. We added the experimental procedure schematic, and moved some panels to supplementary figures.

Reply to the tenth comment made by Reviewer 2: “Mice received 5ug doses of

LNP-mRNA – how does this compare with the current 50- 100ug dose of mRNA COVID vaccines?”

Response:

First, the current COVID-19 vaccine is an antigen-based mRNA vaccine, while this technology is an antibody-based mRNA vaccine. They are two different strategies. Second, mice and humans have different mRNA utilization efficiency and different immune systems. So, it is hard to compare the two cases based on different strategies. More importantly, 5 µg per injection is widely met for mice therapy based on the previous literatures (*Nature* 2016, 534, 396–401; *PNAS* 2022, 119, e2207841119).

Reply to Reviewer 3:

Reply to the first comment made by Reviewer 3: “This paper demonstrates that LNPs with lung-selective mRNA delivery ability produce neutralizing antibodies in the lungs against SARS-CoV-2 infection. Although there have already been papers on LNPs with lung-selective mRNA delivery capability, this paper is valuable in that it uses Lung-LNPs with novel substances and lipid compositions, and demonstrates their therapeutic efficacy against SARS-CoV-2 infection in animals and the safety of the carriers themselves. If the questions in the following points are resolved, the paper may be published in Nature Communication.”

Response:

We thank the highly positive comments from reviewer 3. We revised our manuscript and SI based on these suggestions.

Reply to the second comment made by Reviewer 3: “As shown in Figure 3h, the positive charge of Lung-LNPs is very high (25mV). In the past, cationic liposomes with such a high positive charge had problems with hemolytic properties due to interaction with negatively charged erythrocytes. Therefore, the authors should add the data and discussion by citing previous reports on the interaction with

erythrocytes.”

Response:

We realized the reviewer’s concern, so we carried out the hemolysis test to demonstrate the biocompatibility of lung-LNPs. The results demonstrated that hemolysis was negligible even at higher concentrations (**Extended Data Fig. 13**), demonstrating the safety of the lung-targeting LNPs delivery system. Additionally, we have expanded on this issue with relevant discussions and cited pertinent references.

Reply to the third comment made by Reviewer 3: “The authors should add data and discussion regarding the lipid composition of Lung-LNPs determined in this study for future development of Lung-LNPs research.”

Response:

The corresponding corrections have been done. We added corresponding discussion (lines 422–439), the design strategies were also detailly included to guide future development.

Reviewers' Comments:

Reviewer #2:

Remarks to the Author:

The authors have addressed each of the reviewer requests satisfactorily.

Reviewer #3:

Remarks to the Author:

I think that this revised paper can be acceptable for publication.